# Future Forests: estimating biogenic emissions from net-zero aligned afforestation pathways in the UK

Hazel Mooney[1], Stephen Arnold[1], Ben Silver[1], Piers M Forster[2] and Catherine E Scott[1]

[1]School of Earth and Environment, University of Leeds, LS2 9JT

[2]Priestley Centre for Climate Futures, University of Leeds, LS2 9JT

Correspondence to Hazel Mooney (ee16hm@leeds.ac.uk)

## Abstract

Woodlands sequester carbon dioxide from the atmosphere, which could help mitigate climate change. As part of efforts to reach net-zero greenhouse gas emissions by the year 2050, the UK's Climate Change Committee (CCC) recommend increasing woodland cover from a UK average of 13% to 17-19%. Woodlands can also have benefits for air quality. However, they emit biogenic volatile organic compounds (BVOCs) which are precursors to atmospheric pollutants, such as ozone ($O_3$) and particulate matter (PM), which have the potential to degrade air quality. Here we make an estimate of the potential impact of afforestation in the UK on BVOC emissions, coupling information on tree species' emissions potential, planting suitability and policy-informed land cover change. We quantify the potential emission of BVOCs from five afforestation experiments using the Model of Emissions of Gases and Aerosols from Nature (MEGAN) (v2.1) in the Community Land Model (CLM) (v4.5) for the year 2050. Experiments were designed to explore the impact of variation in BVOC emissions potentials between and within plant functional types (PFTs) on estimates of BVOC emissions from UK land cover, in a future warmer climate under elevated atmospheric $CO_2$ concentrations, to understand the scale of change associated with afforestation to 19% woodland cover by the year 2050.

Our estimate of current annual UK emissions is 39 kt $yr^{-1}$ for isoprene and 46 kt $yr^{-1}$ for total monoterpenes. Broadleaf afforestation results in a change to UK isoprene emission of between -3% and +123%, and a change to total monoterpene emission of between +5% and +48%. Needleleaf afforestation leads to a change in UK isoprene emission of between -3% and +22%, and a change to total monoterpene emission of between +60% and +86%.

Our study highlights the potential for net-zero aligned afforestation, in a likely warmer and drier future UK climate, to have substantial impacts on BVOC emissions. However, the results highlight possible pathways to achieving 19% woodland cover without requiring large increases in isoprene emissions. The emissions estimates presented here provide the opportunity to quantify future impacts on air pollution associated with changes in biogenic emissions, as well as how these impacts would be affected by concurrent changes in anthropogenic emissions.

## 1 Introduction

The terrestrial biosphere plays a key part in suggested pathways for mitigating climate change and helping to reach net-zero greenhouse gas (GHG) emissions, due to the capacity for the biosphere to sequester and store carbon. The Sixth Assessment Report of the Intergovernmental Panel on Climate Change (IPCC) estimated that afforestation, reduced deforestation and ecosystem restoration could mitigate nearly 3 $GtCO_2e$ $yr^{-1}$ globally by 2030 (Shukla et al., 2022). In 2019, the UK Climate Change Committee (CCC) set out a series of pathways to achieving net-zero GHG emissions by 2050, including a large increase in rates of afforestation to expand the land carbon sink (Climate Change Committee, 2019). To achieve their 'further ambition' scenario (a pathway estimated to reduce emissions by 96%), the CCC recommended planting at least 30,000 hectares of trees per year (ha $yr^{-1}$) in the UK until 2050, whilst their 'speculative scenario' (a pathway estimated to reduce emissions by 100% by 2050), recommended increased planting at 50,000 ha $yr^{-1}$ (Climate Change Committee, 2019). Planting at this rate would increase woodland cover from the current UK average of 13% to between 17 and 19%, a relative increase of 31-46%. Following these recommendations, the UK Government committed to increase planting to 30,000 ha $yr^{-1}$ by 2024, and rising from thereafter (Department for Business, Energy and Industrial Strategy, 2021). Planting at this scale requires a doubling of the rate of afforestation in the UK, which has averaged 15,000 ha $yr^{-1}$ between 2019 and 2024 (Forest Research, 2024). Increasing forest cover to such an extent could not only bring benefits for climate change mitigation but also a series of co-benefits including habitat creation, flood risk reduction, improving access for people to trees and woodlands (with the associated  economic and health benefits), reducing concentrations of particulate matter through deposition,  and local temperature reductions (Bolund and Hunhammar, 1999; Costanza et al., 1997; D'Alessandro et al., 2015; Department for Environment, Food and Rural Affairs, 2018; Monger et al., 2022; Nowak, 2022; Purser et al., 2023; Wang et al., 2023). There is also the potential for side-effects and trade-offs, such as the degradation of air quality potentially associated with the emission of biogenic volatile organic compounds (BVOCs) (e.g. Chameides et al., 1988; Churkina et al., 2017; Gu et al., 2021; Rasmussen, 1972).

In addition to influencing atmospheric concentrations of $CO_2$, vegetation exchanges carbon in the form of BVOCs. BVOCs are gaseous compounds synthesised within the chloroplast of the plant, and in some cases, stored in the leaf until volatilised. Examples of BVOC compound classes are isoprene ($C_5H_8$, 2-methyl 1,3-butadiene) and monoterpenes ($C_{10}H_{16}$). Plants use a large amount of energy in the production of BVOCs but appear to derive resilience against pests, disease and other environmental stressors from their emission (Dicke, 2009; Dudareva et al., 2006; Fitzky et al., 2019; Sharkey, 1996). BVOC emission from trees is controlled largely by temperature and light, but also leaf age, atmospheric $CO_2$ concentrations and soil moisture (Guenther et al., 1993; Potosnak et al., 2014; Sharkey, 1996; Zeng et al., 2023). Plant BVOC emissions can be induced by stress from pests and disease; globally the rate at which new tree diseases are reported is doubling approximately every 11 years (Gougherty, 2023), with implications for the quantity and composition of future BVOC emissions (e.g. Dicke, 2009; Ghimire et al., 2022; Irmisch et al., 2014; Jaakkola et al., 2023; Trowbridge and Stoy, 2013). Algorithms have been developed to capture the dependency of BVOCs on the environmental factors mentioned above and to quantify ecosystem- or global-scale emissions of BVOCs (Guenther et al., 1995, 1993, 2012; Martin et al., 2000; Niinemets et al., 1999). Global emissions of BVOCs are estimated at 1000 Tg $yr^{-1}$, of which ~500 Tg $yr^{-1}$ comes from isoprene and ~150 Tg $yr^{-1}$ from total monoterpenes (Guenther et al., 2012). For isoprene, as temperature increases, emissions increase until a maximum temperature ~35 °C after which emissions steeply decline following denaturing of the enzyme isoprene synthase (Monson et al., 1992). Evidence also shows the suppression of isoprene emissions when plants are exposed to elevated atmospheric $CO_2$ concentrations, due to the inhibition of the enzyme responsible for the synthesis of dimethylallyl diphosphate (DMADP) from hydroxymethylbutenyl diphosphate

(HMBDP) (Sahu et al., 2023). The emission of monoterpenes has historically been thought to be
dependent mostly on temperature (Guenther al., 1993), but increasingly evidence shows a light
dependence in monoterpene emission for some plant species (e.g. Bao et al., 2008; Ghirardo et al.,
2010; Jardine et al., 2015). Some plants have a store of monoterpenes (referred to as pool emissions),
which are produced and stored within the leaf until temperatures are high enough for the liquid
compound to volatilise (Tingey et al., 1980). De novo emissions are those synthesised in response to
both light and temperature without a period of storage in the plant; due to the dependence on light,
emissions rates of de novo emissions tend to follow day light patterns (Ghirardo et al., 2010). The
response of monoterpene emissions to climate change is influenced by the behaviour of individual
enantiomers. Byron et al. (2022) found that drought affects monoterpene enantiomers differently,
depending on both the severity and duration of the drought. Since enantiomers can help distinguish
between emissions from de novo synthesis and those from storage pools, differing enantiomeric
responses introduce uncertainty in future emission estimates—especially if the balance between de
novo and storage-derived emissions shifts.
Elevated $CO_2$ has been found to have little effect on monoterpene emissions (Feng et al., 2019). A
global modelling study by Heald et al. (2009) concluded that the enhanced emission of BVOCs
attributed to a warmer climate could be almost entirely offset by $CO_2$ inhibition, despite elevated $CO_2$
enhancing rates of photosynthesis (e.g. Feng et al., 2019; Pegoraro et al., 2004). The climate forcing
impact that a large change in forest cover and associated BVOC emissions may have is uncertain (Scott
et al., 2018; Weber et al., 2024), as is the response of vegetation to elevated $CO_2$ concentrations and
temperature. The point at which temperature or $CO_2$ concentration become inhibiting factors for
vegetation growth and BVOC emission, varies between species and when interacting with other
controls, such as moisture availability (e.g. Fortunati et al., 2008; Pegoraro et al., 2004).

BVOCs are highly reactive and oxidised in the atmosphere by the hydroxyl radical (OH), ozone ($O_3$) and
nitrate radical ($NO_3$). BVOCs influence the production of $O_3$ through a cycle between nitric oxide (NO)
and nitrogen dioxide ($NO_2$), with peroxy radicals derived from BVOC oxidation enabling NO to cycle to
$NO_2$ without depleting $O_3$ and therefore enabling net $O_3$ formation (Sillman et al., 1990; Wennberg et
al., 2018). The oxidation of BVOCs also generates secondary organic aerosol (SOA), which contributes
to the formation of particulate matter (PM) which has negative impacts on human health. Of particular
concern are particles less than 2.5 micrometres in diameter ($PM_{2.5}$) due to the capacity of these
particles to travel further into the human body than larger particles such as $PM_{10}$ (Committee on the
Medical Effects of Air Pollutants, 2015).

Following the rise in ambitions to plant trees for climate change mitigation, several studies have
examined the impact of afforestation on BVOC emissions and air quality (e.g. Gai et al., 2024; Gu et al.,
2021; Purser et al., 2023). Gu et al. (2021) quantified the increase in BVOC emissions from urban
greening scenarios in Los Angeles. They illustrate the potential for the increase in BVOCs to offset the
benefits of a reduction in anthropogenic volatile organic compounds (AVOCs). In the UK context,
Stewart et al. (2003) previously constructed a BVOC inventory for Great Britain, and estimated that
BVOC emissions are 10% of those of AVOCs, though recognising this may change with a warmer
climate.  Using the WRF-EMEP4UK modelling framework, Purser et al. (2023) present an assessment
of the potential air quality impact of BVOCs associated with large-scale afforestation for bioenergy and
short rotation forestry in the UK. Based on four planting scenarios, each of single tree species
(*Eucalyptus gunnii*, *Populus tremulus*, *Alnus cordata* and *Picea sitchensis*) and delivering up to 164%
more woodland cover than present day, Purser et al. (2023) estimated isoprene emissions to increase
between 53% and 135%, except with Alder where emissions declined by 14% due to its lower
emissions potentials relative to the grassland and agriculture being replaced. An increase in
monoterpene emissions between 5% and 94% was simulated, except for Aspen where a decline of 8%
was again attributed to the lower emissions potential of the trees relative to the land cover being
replaced (Purser et al., 2023). To best understand the potential scale of BVOC emissions associated
with afforestation, we need to understand the role that the mixture of tree species planted plays in
determining BVOC emissions.

In this work, we quantify for the first time the emission of BVOCs from UK afforestation scenarios under
which 19% woodland cover is achieved with mixed tree species afforestation. We assume that
afforestation occurs in line with recent and committed afforestation rates for the four UK nations. We
present five afforestation experiments for the UK which represent different pathways to achieving
woodland cover of 19% in 2050, with varying contributions from high and low BVOC emitting tree
species. Our study seeks to explore the potential range of BVOC emissions resulting from a net-zero
consistent level of afforestation, if the afforestation was to occur with feasible but extreme (in terms
of BVOC emission potentials) mixtures of tree species. Estimating potential changes in BVOC emissions
will help guide policy decisions regarding species prioritisation for planting, particularly when
considered in the context of minimising air-quality side-effects associated with afforestation, a concern
noted by The Royal Society in their report on net-zero effects on air quality (2021). The potential impact
that increased emissions of BVOCs could have for air quality (associated with their role as precursors
to atmospheric pollutants) will depend also on the future of anthropogenic emissions and other
precursors, such as $NO_x$, which influence the formation of $O_3$. Our estimates of BVOC emissions from
future afforestation scenarios will facilitate this assessment of future air quality impacts.

## 2 Methods and data
To quantify the emissions of BVOCs from a range of afforestation experiments, we use the Community
Land Model v4.5 (CLM) (Oleson et al., 2013), which includes the  Model of Emissions of Gases and
Aerosols from Nature v2.1 (MEGAN; Guenther et al., 1993, 2012). When operating inside the CLM,
MEGAN takes information about vegetation type and fraction (PFT), LAI and meteorological variables
from the CLM. These are provided as input data when the CLM is operating offline. Within the CLM,
these input variables are used to calculate leaf temperature, soil moisture, soil temperature
and humidity. The CLM also calculates canopy variables, including the fraction of sunlit vs shaded
leaves and the canopy environment coefficient. To calculate BVOC emissions, MEGAN uses leaf
temperature, soil moisture, soil temperature, solar radiation and $CO_2$ concentrations (which are used
to estimate $CO_2$ inhibition of isoprene emission) as well as the canopy environment coefficient
provided by the CLM.
### 2.1 Representing the land surface
We use the CLM (Oleson et al., 2013) at a resolution of 0.47 ° x 0.63 ° with land surface data from the
UKCEH land cover map for 2021 (Marston et al., 2022). The UKCEH map is produced at a resolution of
1 km, combining Sentinel-2 seasonal composite images and 10 context layers to map 21 land cover
classes based on the Biodiversity Action Plan broad habitats (Marston et al., 2022). We use the 1 km
percentage product, which details the percentage cover for each of 21 land cover classes within each
1 km pixel. The UKCEH dataset was regridded to the 0.47 ° x 0.63 ° resolution of our CLM configuration.
As land surface categories vary between the UKCEH and the CLM, variables of the UKCEH map were
reassigned to the closest matching plant functional type (PFT) or land surface category in the CLM,
including the urban, natural vegetation, crops and lake land surface categories. Figure 1 illustrates the
percentage cover of needleleaf and broadleaf tree PFTs and grassland PFTs in the resulting CEH-CLM
dataset which represents present day UK land cover. In this dataset, woodlands cover approximately
13% of UK land, and of this 52% is broadleaf woodland and 48% needleleaf woodland (Forest Research,
2024). Grassland makes up approximately 40% of UK land cover (Marston et al., 2022).

Percentage of land covered by broadleaf, needleleaf and grass PFTs in the UK

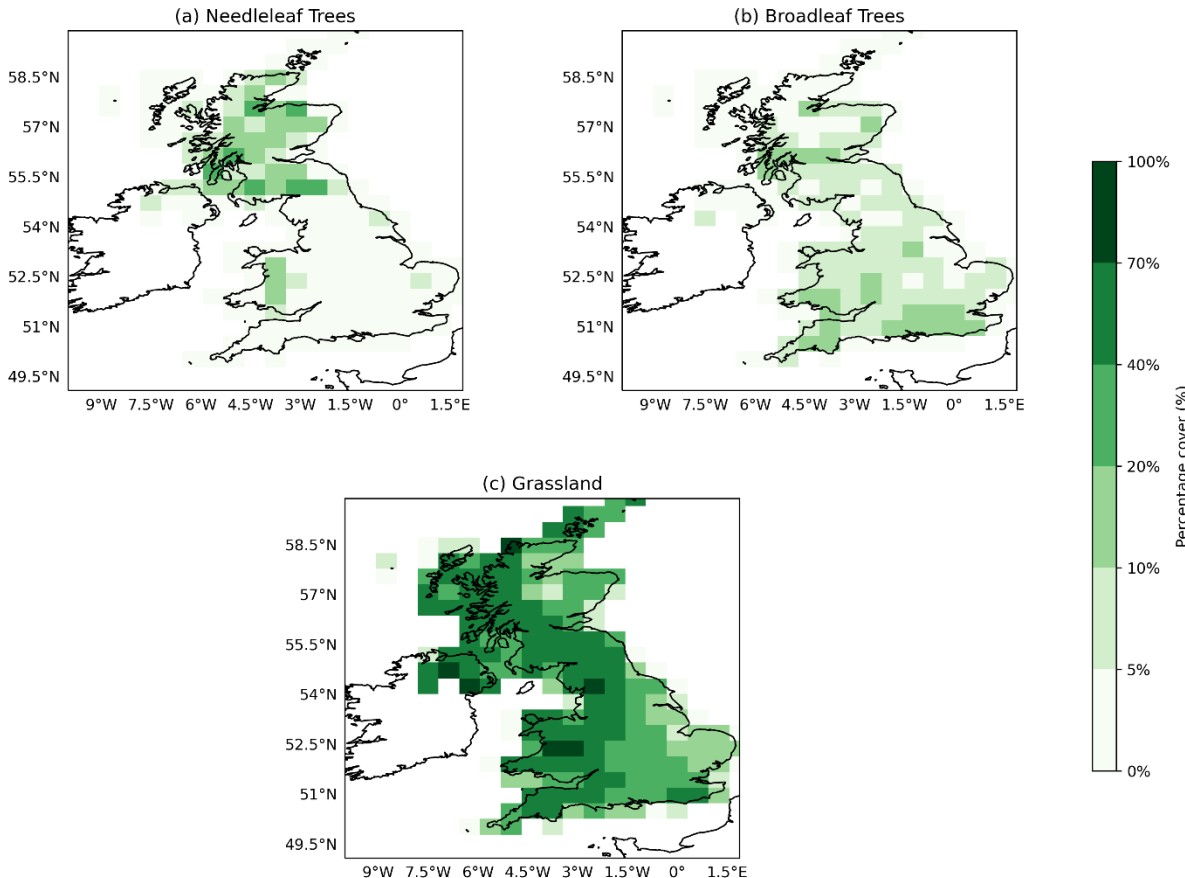


*Figure 1. Percentage share of land in gridcell covered by a) broadleaf trees, b) needleleaf trees, and c) grass PFTs. The*
*distribution of PFTs presented here is inferred from the UKCEH landcover map (1 km x 1 km resolution) (Marston et al., 2022),*
*reassigned to the land cover categories of the CLM (Oleson et al., 2013) to produce the CEH-CLM dataset.*

## 2.2 Identifying land for afforestation

The exact locations for afforestation in the UK are undetermined, though the individual four nations (England, Northern Ireland, Scotland and Wales) have their own ambitions relating to the net-zero aligned planting recommendations (Department for Agriculture, Environment and Rural Affairs, 2020; Department for Environment, Food and Rural Affairs, 2021; Scottish Government, 2017 and Welsh Government, 2018). For our study, we present five afforestation experiments equivalent to the achievement of 19% woodland cover by 2050 in the UK. We assume a direct replacement of grassland with woodland as a simplified land cover change. In reality, detailed assessments are required to establish the suitability (ecologically, socially and culturally) of any site for woodland creation; conversion to woodland in our study is used only to estimate the impact on BVOC emissions and should not be used to infer the suitability of any specific site.

Whilst a lack of specific locations for future additional woodland adds extra uncertainty to any estimates of future BVOC emissions, we seek to minimise this by distributing the new woodland in accordance with historical planting rates and future planting commitments across the four nations of the UK (Department for Agriculture, Environment and Rural Affairs, 2020; Department for Environment, Food and Rural Affairs, 2021; Forest Research, 2023; Scottish Government, 2017 and Welsh Government, 2018). Table 1 details the rates of afforestation reported for each nation during the past five decades, their current commitments to afforestation and the proposed share of afforestation implemented in this study. Our experiments assume the greater level of afforestation from the CCC's recommendations (equivalent to planting 50,000 ha yr$^{-1}$), as this enables estimation of the change in BVOCs associated with the greatest scale of afforestation in recent discourse for the UK. We assume 63% of the UK's new woodland is created in Scotland, 8% in Wales, 4% in Northern Ireland and 25% in England.

*Table 1. Historical rates of afforestation in the UK by nation between 1976 and 2021 and current national ambitions for*
*afforestation. These rates inform the share of UK afforestation at 50,000 ha yr$^{-1}$ used in this study.*

|  | Scotland | Wales | Northern Ireland | England |
|---|---|---|---|---|
| Mean share of UK afforestation 1976-2021 (Forest Research, 2023) | 70% | 4% | 4% | 22% |
| Most recent statement on afforestation ambitions (equivalent share of UK total given as percentage in brackets) | 15,000 ha yr$^{-1}$ by 2024 (63%) (Scottish Government, 2017) | 2000 ha yr$^{-1}$ from 2020 onwards (8%) (Welsh Government, 2018) | 900 ha yr$^{-1}$ from 2020 to 2030 (4%) (Department for Agriculture, Environment and Rural Affairs, 2020) | Trebling current rates (increasing from around 2000 to 6000 ha yr$^{-1}$) (25%) (Department for Environment, Food and Rural Affairs, 2021) |
| This study: share of UK afforestation at 50,000 ha yr$^{-1}$ | 31,500 ha yr$^{-1}$ (63%) | 4000 ha yr$^{-1}$ (8%) | 2000 ha yr$^{-1}$ (4%) | 12,500 ha yr$^{-1}$ (25%) |

We use the presence of a tree PFT within a grid cell of the CEH-CLM dataset as indicative of climatic suitability of the land for new planting of that type. The tree PFTs considered in this study are Broadleaf Deciduous Temperate Trees (BDTT), Broadleaf Deciduous Boreal Trees (BDBT), Needleleaf Evergreen Temperate Trees (NETT) and Needleleaf Evergreen Boreal Trees (NEBT). For replacement with woodland to take place, grid cells are required to have some existing grassland (namely Cool C3 or Arctic C3 grass). New tree cover was created in proportion to the amount of grassland in any eligible grid cell and in the same ratio of temperate to boreal PFTs as was already present in the CEH-CLM dataset. Afforestation was implemented as either all needleleaf, or all broadleaf, to enable examination of the variation in emissions that would be generated by different planting decisions. This method does not account for the specific suitability of a given location for tree planting but provides the necessary scope to consider the scale of emissions changes across the UK associated with present ambitions for afforestation. Figure 2 shows the absolute change in percentage area of PFTs following afforestation. The quantities of change in PFT cover are consistent across both needleleaf and broadleaf afforestation experiments. Supplementary Fig. S1 and S2 illustrate the percentage change in area of PFTs following broadleaf and needleleaf afforestation respectively.

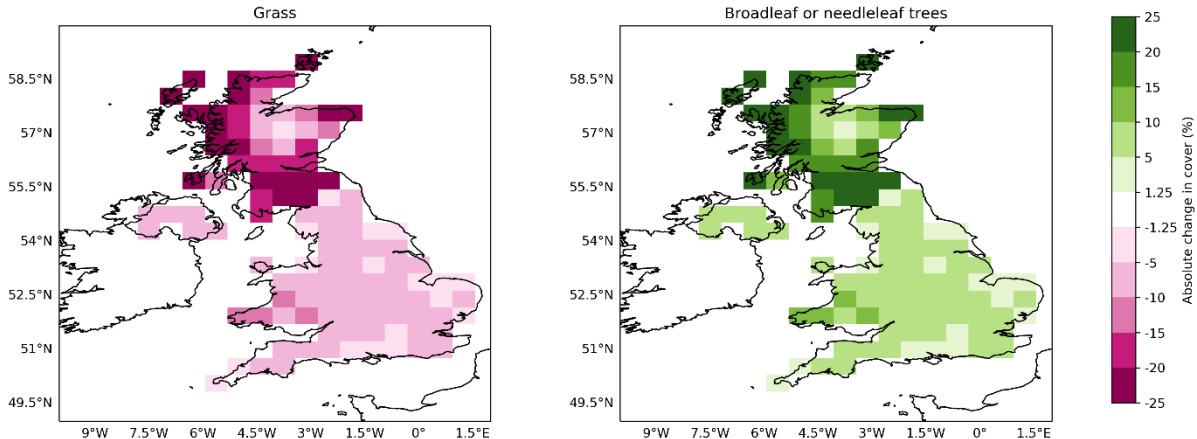

*Figure 2. Absolute change in the percentage cover of grass and tree PFTs following either needleleaf or broadleaf afforestation.*


## 2.3 Identifying tree species for afforestation

The subset of tree species represented in this study was informed by the Forest Research tree species
database (Forest Research, 2024a) and associated Ecological Site Classification (ESC) tool. This online
tool enables users to identify suitable trees for planting in each part of the UK according to known or
projected conditions for climate, lithology, ecology, soil moisture and soil nutrients (Pyatt et al., 2001).
60 tree species are considered in the ESC tool. Of these species, those classified as principal species
(defined as species already widely used or increasing in deployment in the UK, and of continuing
importance unless adversely affected by a new pest or disease, or climate change) (Tree Species
Database, 2024) were considered for representation in our study. We compare this subset of 25 tree
species to statistics on current tree species composition in the UK (Forest Research, 2023) and the EU-
Trees4F project which examines the changing distribution of European tree species under a selection
of climate futures considering natural dispersal and planting (Mauri et al., 2022). Based on these
requirements, we selected 18 tree species to include in our review of emissions potentials data (details
of included tree species can be found in Supplementary Table S1).

## 2.4 Preparing UK specific emissions potential data

We generate a bespoke UK-specific emissions potential dataset to capture isoprene and monoterpene
emissions from UK-specific tree species. We use this dataset to adjust estimates of BVOC emission
from tree species for our present-day land cover and afforestation experiments. Tree species specific
emissions potentials were used to generate PFT emissions potential scenarios based on both present-
day tree species abundance in the UK, and a range of hypothetical planting mixtures that demonstrate
the variation within and between PFTs. We reviewed European or UK specific emissions potentials of
tree species to identify values representative of UK trees. Emissions data were obtained for isoprene
and monoterpene compound classes, as the BVOCs dominating VOC chemistry in Great Britain
(Atkinson, 1990). If data was available for tree species in our subset, emissions values were recorded
(applicable studies are summarised in Table 2). The review returned emissions potential data for
isoprene and monoterpenes for 6 needleleaf and 10 broadleaf tree species (listed in Table 3; additional
details can be found in Supplementary Table S1) which make up existing UK woodland cover and/or
are expected to play a role in afforestation in the future.

Monoterpenes have pool emissions (previously synthesised compounds emitted from storage pools)
as well as synthesis (or *de novo;* compounds synthesised and emitted in response to light) emissions.
The review of emissions potential literature highlighted inconsistencies in the level of differentiation
between pool and synthesis emissions. In MEGAN, the relative light dependence of BVOC compound
class emissions is managed by assigning a light dependent fraction (Guenther et al., 2012). We account
for this by taking the sum, where synthesis and pool emission potentials are explicit in the published
dataset, and reapplying the existing light dependent fractions in MEGAN.
The emissions potentials values obtained from different studies were used to calculate a mean
emission potential for isoprene and total monoterpenes for each tree species. Figures 3 and 4 show
the relationship between isoprene and total monoterpene emissions potentials for broadleaf (Fig. 3)
and needleleaf (Fig. 4) tree species and the emissions potential of corresponding PFTs from MEGAN
(namely BDTT and BDBT for broadleaf species and NETT and NEBT for needleleaf species) (Guenther
et al., 2012). MEGAN provides values as emissions factors. To enable comparison, we converted
emissions factors from MEGAN (Guenther et al., 2012) to emissions potentials using the Eq 1., where
$E_f$ = emissions factor (µg m$^{-2}$ hr$^{-1}$), $E_p$ = emissions potential (µgC gDW$^{-1}$ hr$^{-1}$) and $F_d$ = foliar density, given
as 560 gDW m$^{-2}$ for deciduous trees and 1100 gDW m$^{-2}$ for conifer trees (Guenther et al., 1995).
*Equation 1*

$$E_p = \frac{E_f \left(\frac{60}{68}\right)}{F_d}$$

*Table 2. Overview of tree emissions potential studies reviewed, including the number of tree species that data were presented*
*for, and the number of tree species incorporated into this study.*

| Study number | Study | Study region | Number of tree species presented | Tree species from the study incorporated in this work |
|---|---|---|---|---|
| 1 | Simpson *et al.,* 1999 | Europe | 37 | 15 |
| 2 | Hewitt, 2003 | Great Britain | 1100 | 10 |
| 3 | Stewart *et al.,* 2003 | Great Britain | 1100 | 5 |
| 4 | Karl *et al.,* 2009 | Europe | 112 | 15 |
| 5 | Churkina *et al.*, 2017 | Berlin | 11 | 6 |
| 6 | Purser *et al.,* 2021 | United Kingdom | 3 | 1 |
| 7 | Purser *et al.,* 2023 | United Kingdom | 4 | 2 |

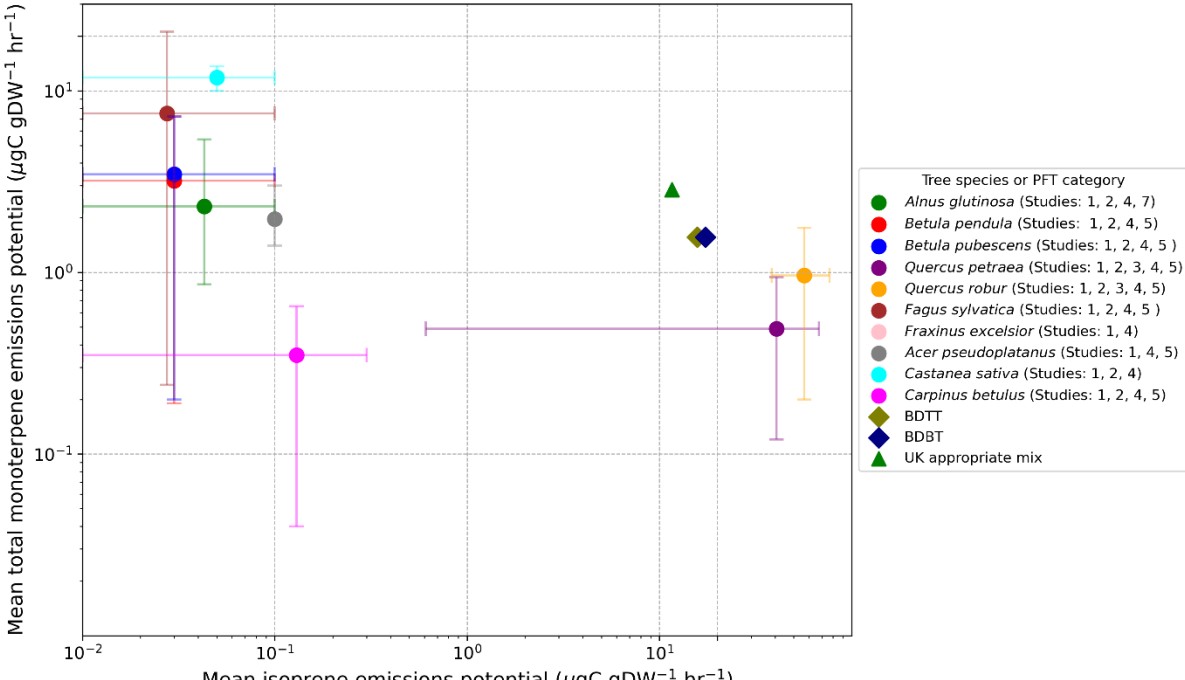


Figure 3. Relationship between the mean isoprene and mean total monoterpene emissions potentials for UK appropriate broadleaf tree species. Circular markers represent the mean of values obtained for a given broadleaf tree species from the literature (see Table 2). Triangular markers represent a UK appropriate value calculated as a mean of broadleaf species emissions potentials weighted according to their present-day abundance in the UK (Forest Research, 2023). Diamond markers represent the emissions potentials of corresponding PFT categories within the default release version of MEGANv2.1 (BDTT and BDBT). Numbers in the figure legend correspond to the study number detailed in Table 2. Error bars indicate the minimum and maximum values obtained from the literature.

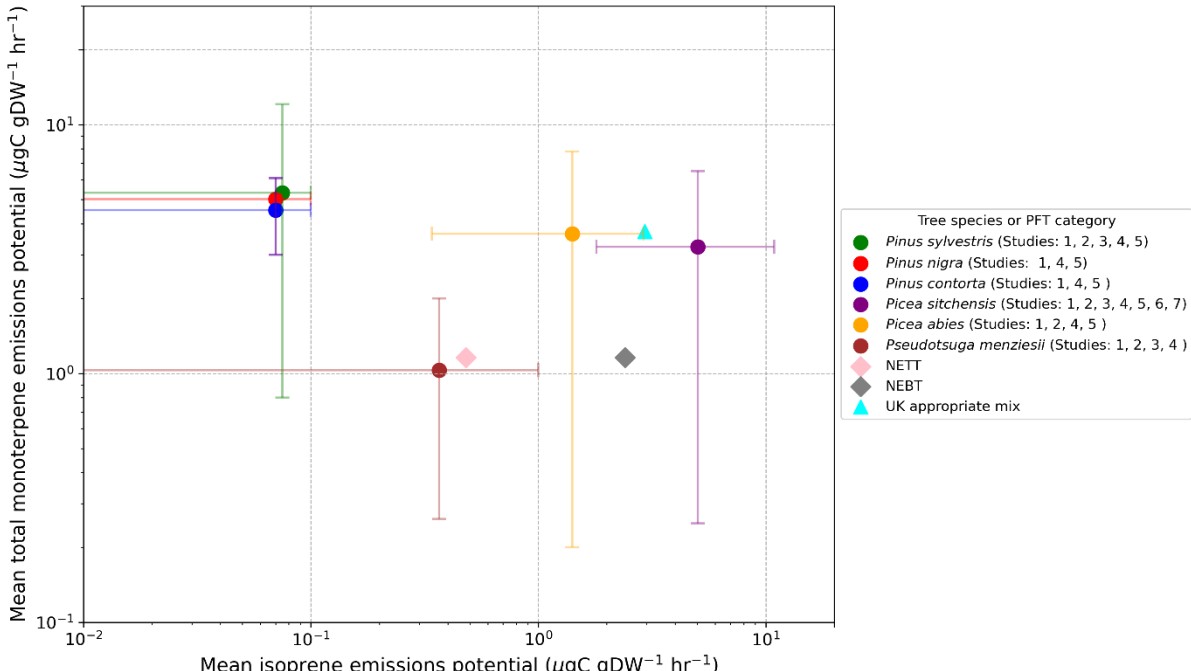

*Figure 4. Relationship between the mean isoprene and mean total monoterpene emissions potentials for UK appropriate needleleaf tree species. Circular markers represent the mean of values obtained for a given needleleaf tree species from the literature (see Table 2). Triangular markers represent a UK appropriate value calculated as a mean of needleleaf species emissions potentials weighted according to their present-day abundance in the UK (Forest Research, 2023). Diamond markers represent the emissions potentials of corresponding PFT categories within the default release version of MEGANv2.1 (NETT and NEBT). Numbers in the figure legend correspond to the study number detailed in Table 2. Error bars indicate the minimum and maximum values obtained from the literature.*

In the case of broadleaf species (Fig. 3), emissions potentials suggest three broad categories, where higher emitters of monoterpenes are generally lower emitters of isoprene (e.g. *Fagus sylvatica* and *Betula pendula*), higher emitters of isoprene are generally lower emitters of monoterpenes (e.g. *Quercus spp.*) and some are low emitters of both (e.g. *Fraxinus excelsior*). To reflect these groupings, we designed two broadleaf emissions scenarios, *BL_highMono_lowIso* and *BL_lowMono_highIso* (described in Table 3). Needleleaf tree species (Fig. 4) seem to be generally high emitters of monoterpenes, but either lower emitters of isoprene (e.g. *Pinus spp.*) or higher emitters of isoprene (e.g. *Picea spp.*). To reflect these groupings, we designed two needleleaf emissions scenarios: *NL_highMono_lowIso* and *NL_highMono_highIso*. The emission potentials used in each scenario are calculated as a mean of the tree species in that group, based on the mean of values presented in the literature. We developed a fifth emissions scenario, *UK_appropriate_mix*, with values calculated as a mean of species emission factors weighted according to their present-day abundance in the UK, based on the 2023 Woodland Statistics (Forest Research, 2023). A scenario using the existing emissions factors for these PFTs in MEGAN is referred to as the *Default_MEGAN* scenario. Emissions potentials for each scenario are illustrated in Fig. 5. For use in MEGAN, emissions potentials are converted to emissions factors. The emissions factors of each scenario are detailed in Table 3 for isoprene and total monoterpenes. For use in MEGAN, values for total monoterpene emissions factors were allocated to compound classes according to the ratio of those classes in Default MEGAN emissions factors. Emissions factors are combined with information on vegetation type and distribution (PFTs), LAI, solar radiation and atmospheric $CO_2$ concentration and (calculated by the CLM) soil moisture, soil temperature, leaf temperature and the canopy environment coefficient for calculating emissions in MEGAN.

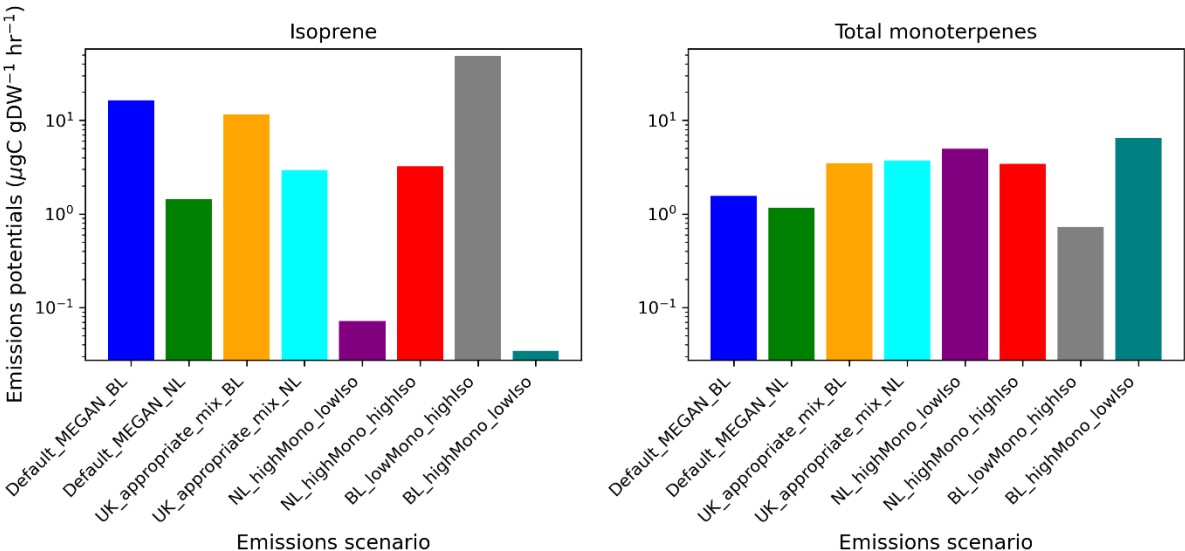

*Figure 5. Emissions potentials for isoprene and total monoterpene compound classes, across the emissions scenarios described in Table 3. For default MEGAN and UK appropriate mix scenarios, values are provided for both broadleaf (BL) and needleleaf (NL). For all other scenarios, when broadleaf or needleleaf is unspecified, UK appropriate mix values have been used.*
*Table 3. Emission scenario short name and description, related PFT, example representative UK tree species informing the*
*scenario, and emissions factors by BVOC compound class for isoprene and total monoterpene. The emissions scenarios are:*
*Default_MEGAN, UK_appropriate_mix, NL_highMono_lowIso, NL_highMono_highIso, BL_lowMono_highIso and*
*BL_highMono_lowIso. Where PFT is specified as broadleaf, this includes BDTT and BDBT PFTs. Where PFT is specified as*
*needleleaf, this includes NETT and NEBT PFTs.*

| Emission scenario short name and description | PFT | Species | Emissions Factors ($\mu g\ m^{-2}\ hr^{-1}$) | |
|---|---|---|---|---|
| | | | **Isoprene** | **Total monoterpenes** |
| *Default MEGAN:* Emissions factors as provided in MEGANv2.1 default set up | BDBT | Unspecified | 7000 | 1300 |
| | BDTT | Unspecified | 10,000 | 990 |
| | NEBT | Unspecified | 3000 | 1450 |
| | NETT | Unspecified | 600 | 1450 |
| *UK_appropriate_mix:* Emissions factors represent present-day abundance of dominant tree species | Broadleaf | *Alnus glutinosa, Betula pendula, Betula pubescens, Quercus petraea, Quercus robur, Fagus sylvatica, Fraxinus excelsior, Acer pseudoplatanus, Castanea sativa, Carpinus betula* | 7359 | 1808 |
| | Needleleaf | *Pinus sylvestris, Pinus nigra, Pinus contorta, Picea sitchensis, Picea abies, Pseudotsuga menziesi* | 3660 | 4643 |
| *NL_highMono_lowIso:* Mean emissions factors from high monoterpene - low isoprene emitting needleleaf trees | Broadleaf | UK appropriate mix | 7359 | 1808 |
| | Needleleaf | *Pinus sylvestris, Pinus nigra, Pinus contorta* | 89 | 6188 |
| *NL_highMono_highIso*: Mean emissions factors from high monoterpene - high isoprene emitting needleleaf trees | Broadleaf | UK appropriate mix | 7359 | 1808 |
| | Needleleaf | *Picea sitchensis, Picea abies* | 4014 | 4282 |
| *BL_low_Mono_highIso:* Mean emissions factors from low monoterpene - high isoprene emitting broadleaf trees | Broadleaf | *Quercus robur, Quercus petraea* | 30892 | 460 |
| | Needleleaf | UK appropriate mix | 3660 | 4643 |
| *BL_highMono_lowIso*: Mean emissions factors from high monoterpene - low isoprene emitting broadleaf trees | Broadleaf | *Castanea sativa, Fagus sylvatica, Betula pubescens, Betula pendula* | 22 | 4124 |
| | Needleleaf | UK appropriate mix | 3660 | 4643 |


## 2.5 Model set up

Simulations were undertaken at 0.47 ° x 0.63 ° resolution with the CLM component of Community Earth System Model (CESM) version 2.2.0 (Danabasoglu et al., 2020; Oleson et al., 2013). BVOC emissions are calculated by MEGAN (Guenther et al., 1993; 2012). Land surface input data for CLM was modified to reflect the CEH-LCM distribution of PFTs over the UK, generating the land surface dataset CEH-CLM used in this study, as detailed in Sec. 2.1. Meteorological data is from the Global Soil Wetness Project (see Dirmeyer *et al.*, (2006)), for the year 2003. Values of leaf area index (LAI) are those developed for the CLM from the 1 km MODIS product as detailed in Lawrence and Chase (2007). Detailed information on model component sets for CESM2.2.0 is available from NCAR (https://docs.cesm.ucar.edu/models/cesm2/config/compsets.html). An initial experiment to estimate present-day BVOC emissions uses an atmospheric $CO_2$ concentration of 375 ppm, appropriate for the year 2003. We modify input $CO_2$ concentration data in all other simulations to 500 ppm based on the projected atmospheric $CO_2$ concentration in 2050 under SSP2-4.5 (Intergovernmental Panel on Climate Change, 2023). Previous studies have explored the impact of drought on BVOC emissions (e.g. Potosnak et al., 2014) and developed parameterisations to improve the representation of this process in MEGAN (e.g. Jiang et al., 2018; Wang et al., 2022). To enable quantification of the sensitivity of our results to the representation of the impact of drought on isoprene emissions, extra simulations were run with a modification to the soil moisture activity factor algorithm following Wang et al., (2022), see Supplementary Table S2.

## 2.6 Meteorology

The controls on BVOC emission include temperature, atmospheric $CO_2$ concentration and soil moisture (Guenther et al., 1993; Sharkey, 1996). To determine the appropriate meteorological data for simulating likely conditions in the year 2050, we compared observed and projected UK temperatures from the UK Climate Projections (UKCP) project (UKCP, 2023a, b). Based on examination of HadUK-Grid daily observations of maximum air temperature at 1.5 m averaged for the UK for the years 2000-2021 (UKCP, 2023b) and global projections of daily maximum temperatures at 1.5 m for the years 2045-2054 (UKCP, 2023a), we determined that meteorology for the year 2003 was the best representation of projected 2050 temperatures in the historical dataset. The heatwave periods of the year 2003 have previously been studied for impacts on air quality, so this period provides the opportunity for comparison of modelled data to observations (Lee et al., 2006; Stedman, 2004; Vautard et al., 2005).

## 2.7 Model simulations

Table 4 describes the configurations used in 12 experiments to explore changes in UK BVOC emissions with and without additional forests. We carry out seven experiments without any additional forest, and five with afforestation aligned with achievement of approximately 19% UK woodland coverage. At this resolution, the resulting level of afforestation in model experiments from which emissions are estimate in equivalent to the achievement of 18.65% woodland cover by the year 2050.

*Table 4. Summary of model run configurations: afforestation experiment short name, experiment description and input data: land cover, emission scenario short name and associated emission factors and atmospheric $CO_2$ concentration data. In experiment names 'Present_day…' refers to the land cover. The latter half of the name refers to the emissions potentials.*

| Afforestation experiment short name | Experiment description | Land data | Emission scenario short name | $CO_2$ concentration |
|---|---|---|---|---|
| Baseline | Present day land cover with UK appropriate emission factors and 2003 $CO_2$ | CEH-CLM (13% woodland cover) | UK_appropriate_mix | 375 ppm |
| Present_day_Default_MEGAN | Present day land cover with default MEGAN emissions factors and elevated $CO_2$ | CEH-CLM (13% woodland cover) | Default_MEGAN | 500 ppm |
| Present_day_Current_UK_mix | Present day land cover with UK appropriate emission factors and elevated $CO_2$ | CEH-CLM (13% woodland cover) | UK_appropriate_mix | 500 ppm |
| Present_day_NL_highMono_lowIso | Present day land cover with high monoterpene emission factors from needleleaf trees and elevated $CO_2$ | CEH-CLM (13% woodland cover) | NL_highMono_lowIso | 500 ppm |
| Present_day_NL_highMono_highIso | Present day land cover with high monoterpene and isoprene emission factors from needleleaf | CEH-CLM (13% woodland cover) | NL_highMono_highIso | 500 ppm |

| | | | | |
|---|---|---|---|---|
| | trees and elevated $CO_2$ | | | |
| Present_day_BL_lowMono_highIso | Present day land cover with high isoprene emission factors from broadleaf trees and elevated $CO_2$ | CEH-CLM (13% woodland cover) | BL_lowMono_highIso | 500 ppm |
| Present_day_BL_highMono_lowIso | Present day land cover with high monoterpene emission factors from broadleaf trees and elevated $CO_2$ | CEH-CLM (13% woodland cover) | BL_highMono_lowIso | 500 ppm |
| Afforested_Current_UK_mix | Afforested land surface with UK appropriate emission factors from broadleaf and needleleaf trees assuming planting in the current UK mix, and elevated $CO_2$ | CEH-CLM afforested to 18.65% woodland cover (current UK ratio of broadleaf to needleleaf). | UK_appropriate_mix | 500 ppm |
| Afforested_BL_lowMono_highIso | Afforested land surface with new broadleaf trees of high isoprene emission factors and elevated $CO_2$ | CEH-CLM afforested to 18.65% woodland cover (100% broadleaf) | BL_lowMono_highIso | 500 ppm |

| | | | | |
|---|---|---|---|---|
| Afforested_BL_highMono_lowIso | Afforested land surface with new broadleaf trees of high monoterpene emission factors and elevated $CO_2$ | CEH-CLM afforested to 18.65% woodland cover (100% broadleaf) | BL_highMono_lowIso | 500 ppm |
| Afforested_NL_highMono_lowIso | Afforested land surface with new needleleaf trees of high monoterpene emission factors and elevated $CO_2$ | CEH-CLM afforested to 18.65% woodland cover (100% needleleaf) | NL_highMono_lowIso | 500 ppm |
| Afforested_NL_highMono_highIso | Afforested land surface with new needleleaf trees of high monoterpene and isoprene emission factors and elevated $CO_2$ | CEH-CLM afforested to 18.65% woodland cover (100% needleleaf) | NL_highMono_highIso | 500 ppm |


3 Results and Discussion
3.1 Estimating present-day annual UK emissions of isoprene and total monoterpenes
Figure 6 illustrates the distribution of annual UK emissions for isoprene and total monoterpene
compound classes from our *Baseline* experiment (Table 4). We present an estimate of present-day UK
annual isoprene emission of 39 kt yr$^{-1}$ and total monoterpene emission of 46 kt yr$^{-1}$. Our estimates of
BVOC emissions with present-day land cover fall within the range of values presented in previous
studies for the UK or Great Britain, i.e. 8 - 110 kt yr$^{-1}$ for isoprene and 31 - 145 kt yr$^{-1}$ for total
monoterpenes (Guenther et al., 1995; Hayman et al., 2017; Purser et al., 2023; Simpson et al., 1999;
Stewart et al., 2003). The spatial distribution of emissions is similar to that illustrated in previous UK
studies, such as Purser et al. (2023) and Stewart et al. (2003), with the greatest isoprene emission rates
located in the southern-most parts of England and around the Scotland-England border, whilst
monoterpene emissions are concentrated in Scotland, and parts of Wales.

Modelled annual emissions of isoprene and monoterpenes from present day land cover

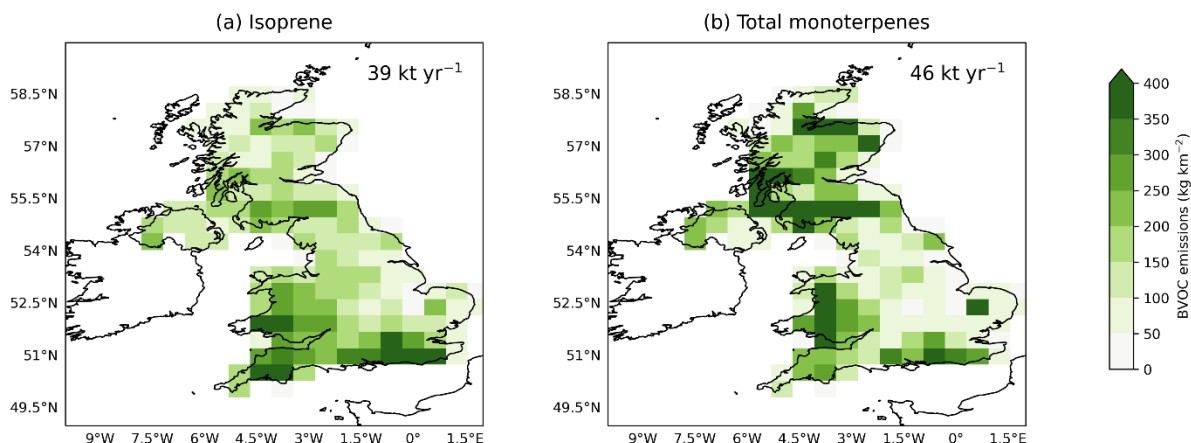


*Figure 6. Modelled distribution of emissions of isoprene (kg km$^{-2}$) and total monoterpene (kg km$^{-2}$) from present day UK land*
*cover. Figures 6a and 6b show the emissions generated from the Baseline experiment. The UK total (thousand tonnes) is given*
*in the text overlay.*
3.2 Investigating the impact of emissions factors and $CO_2$ concentration
Figure 7 illustrates the distribution of annual UK emissions for isoprene and total monoterpenes from
present day land cover for experiments *Present_day_Current_UK_mix* (Fig. 7a and 7b) and
*Present_day_Default_MEGAN* (Fig. 7c and 7d).  In these experiments $CO_2$ concentrations are elevated
to those projected for the year 2050 to enable calculation of the estimated change in emissions
attributed to afforestation only in afforested experiments (Sect. 3.3). Experiment
*Present_day_Current_UK_mix* simulates isoprene emissions lower than our *Baseline* experiment, at
33 kt yr$^{-1}$, whilst emissions of total monoterpene are unchanged. The decline of 15% in isoprene
emissions, in the absence of any change to woodland coverage, can be attributed to the $CO_2$ inhibition
effect on the rate of isoprene emission from trees.
Estimates of BVOCs have been made on the global scale and continental scale (Guenther et al., 1995;
Simpson et al., 1999), but few estimates have been made for the UK (e.g. Purser et al., 2023; Simpson
et al., 1999; Stewart et al., 2003). Further, studies have demonstrated the impact of providing localised
tree species information on estimates of BVOCs for a given location, compared to coarser resolution
forest data. For example, Luttkus et al. (2022) examined the impact of detail in land use data on air
quality predictions, showing substantial differences between estimates of BVOC emissions, and their
distribution, when detail on tree species distribution was improved.  The review of emissions potential
data presented in Sect. 2.4 illustrated that emissions potentials of isoprene from broadleaf PFTs in our
*UK_appropriate_mix* scenario are lower than those in the *Default_MEGAN* scenario, whilst values for
needleleaf PFTs were slightly higher in the *UK_appropriate_mix* scenario than those in the
*Default_MEGAN* scenario (Fig. 5). Combined, these variations result in the small decrease of 6% in
isoprene overall in our experiment (to 33 kt yr$^{-1}$) when emissions data is adapted to reflect UK tree
species (Fig. 7). In contrast, the emissions potentials data found values for monoterpenes in both
broadleaf and needleleaf PFTs to be largely underestimated in MEGAN when compared to a range of
UK tree species (Fig. 7). This is reflected in the estimate of total monoterpene emissions from our
*Present_day_Current_UK_mix* experiment (at 46 kt yr$^{-1}$) being more than twice that of our
*Present_day_Default_MEGAN* experiment. Adjusting emissions factors to represent the current
relative abundance of tree species in the UK delivers approximately 142% extra monoterpene
emissions. This demonstrates the importance of preparing bespoke input data for examining BVOC
emissions, where regionally appropriate emissions factors can be applied.

Modelled annual emissions of isoprene and monoterpenes from present day land cover with elevated $CO_2$

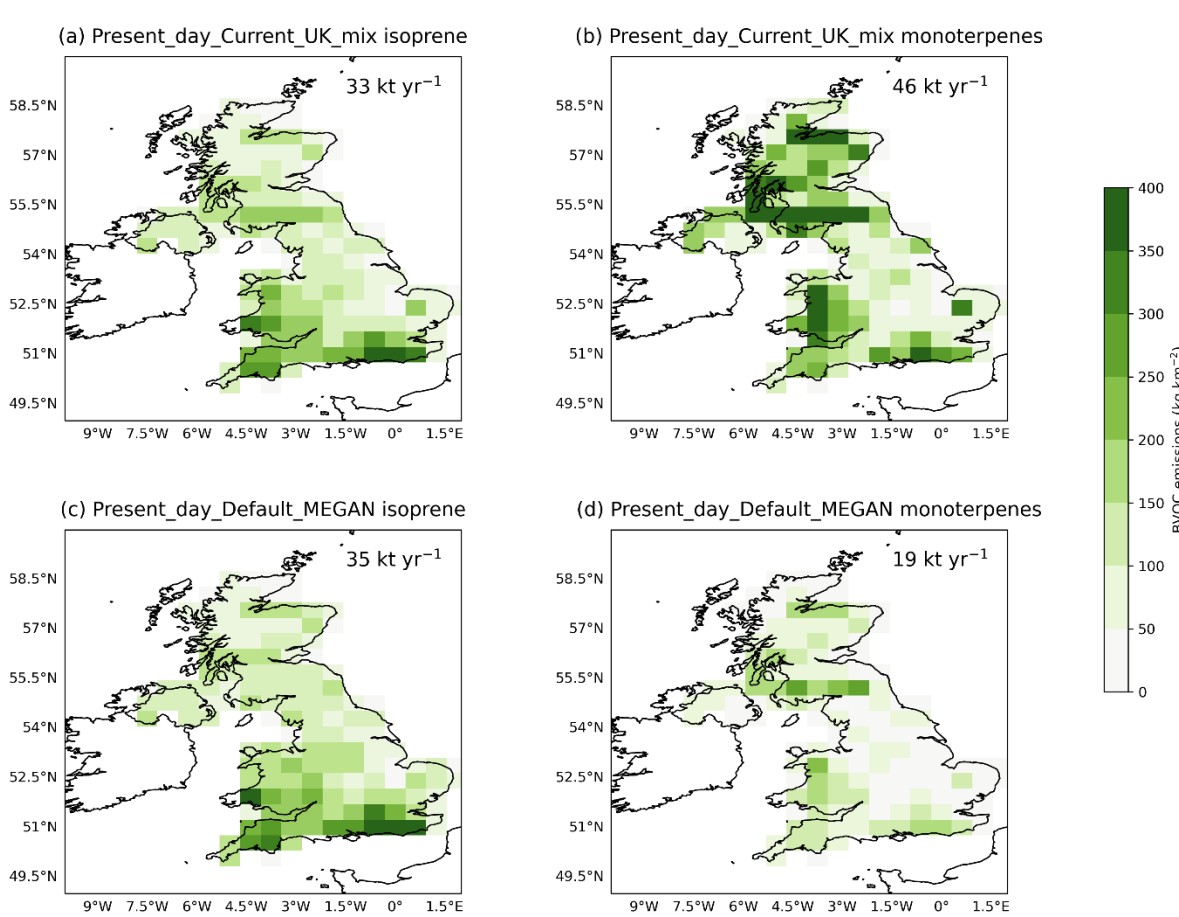


*Figure 7. Modelled distribution of emissions of isoprene (kg km$^{-2}$) and total monoterpene (kg km$^{-2}$) from present day UK land cover. Figures 7a and 7b show the emissions generated from the Present_day_UK_appropriate_mix experiment. Figures 7c and 7d show the emissions generated with the Present_day_Default_MEGAN experiment. The UK total (thousand tonnes) is given in the text overlay.*

3.3 Estimating BVOC emissions attributed to UK afforestation
Figure 8 illustrates the percentage change in annual UK emissions of isoprene and total monoterpenes
across the five afforestation experiments relative to present day land cover. All experiments used to
estimate percentage change in emissions use elevated $CO_2$ as projected for 2050, therefore all
percentage changes in emissions are attributed to afforestation alone. Modelled estimates of total
annual UK emissions of isoprene and total monoterpene compound classes are given in the text
overlay. Values of percentage change in annual UK emissions and total annual estimates are also
summarised in Table 5.

Percentage change in annual emissions of isoprene and total monoterpenes following afforestation

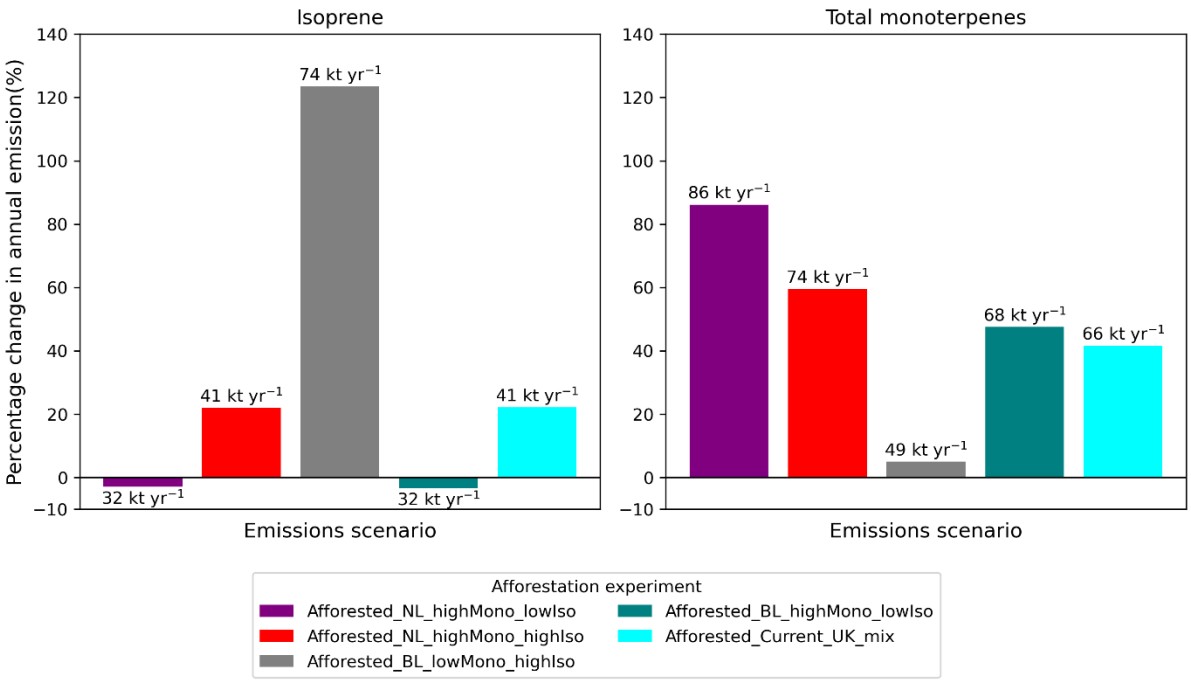


*Figure 8. Percentage change in modelled annual UK emissions of isoprene and total monoterpene following afforestation to*
*19% woodland cover for five afforestation experiments: Afforested_BL_lowMono_highIso, Afforested_BL_highMono_lowIso,*
*Afforested_NL_highMono_lowIso, Afforested_NL_highMono_highIso and Afforested_Current_UK_mix as detailed in Table 6.*
*The total annual UK emissions of isoprene and total monoterpenes of each experiment are given in the text overlay. The*
*percentage change reported here is relative to our Present_day_Current_UK_mix experiment, with the change attributed to*
*the different in forest cover (all experiments using elevated $CO_2$ at 500 ppm).*
In all scenarios, the larger increases in emissions are concentrated in Scotland (see Supplementary Fig.
S3 and S4 for maps of percentage change in emissions, due to the scale of planting introduced there
relative to the other UK nations (Table 1). The greatest increase in isoprene was observed in the
experiment *Afforested_BL_lowMono_highIso*, where individual grid cells experienced up to a 250%
increase in emissions, and total emissions increased by 123%. In experiments
*Afforested_BL_highMono_lowIso* and *Afforested NL_highMono_lowIso*, the replacement of grassland
with tree cover results in a reduction of isoprene emissions, of -3%, and as much as -50% on an
individual grid cell level. This is attributed to the lower emission factors applied to tree PFTs relative
to those representing grass PFTs. The adaptations to prepare bespoke UK appropriate emissions
potential values were not also made for non-tree PFTs. Therefore, our study may not present the best
possible representation of UK non-tree PFTs, and so the decline in emissions attributed to
replacement of grassland with woodland may not accurately reflect the case of UK grasses. The
potential change in BVOC emissions from afforestation would vary with the type of land cover
replaced, bringing some uncertainty to the direction of change in emissions in the future. However,
similar outcomes were found by Purser et al. (2023) where afforestation was focused on short rotation

forestry, with a decline in isoprene emissions of 14% when examining the impact of large-scale planting of alder. In Fig. 8, we see an increase in monoterpene emissions under all scenarios. The greatest changes are observed in the *Afforested_NL_highMono_lowIso* scenario where some grid cells experience an increase above 250%, and total monoterpene emissions increased by 86%. Experiments *Afforested_NL_highMono_highIso*, *Afforested_BL_highMono_lowIso* and *Afforested_Current_UK_mix* demonstrate moderate increases in monoterpene emissions, between 42 and 60%. The smallest change in monoterpene emissions is observed in our *Afforested_BL_lowMono_highIso* experiment, where a 5% increase is simulated.

Our modelled estimates of isoprene emission for the UK with 19% woodland coverage vary between 32 kt yr$^{-1}$ and 74 kt yr$^{-1}$ across the experiments; modelled estimates of total monoterpene emission are between 49 kt yr$^{-1}$ and 86 kt yr$^{-1}$. Overall, our results indicate that emission factors and tree species mixture substantially change estimates of BVOC emissions from the UK's future forests.

*Table 5. Modelled estimates of annual UK isoprene and total monoterpene emissions (thousand tonnes) and percentage change (relative to Present_day_UK_appropriate mix) following afforestation to 19% woodland cover for five afforestation experiments.*

| Afforestation experiment | Estimated annual emission (kt species yr$^{-1}$) | |
|---|---|---|
| | Isoprene | Total monoterpenes |
| Afforested_Current_UK_mix | 41 (+22%) | 66 (+42%) |
| Afforested_BL_lowMono_highIso | 74 (+123%) | 49 (+5%) |
| Afforested_BL_highMono_lowIso | 32 (-3%) | 68 (+48%) |
| Afforested_NL_highMono_lowIso | 32 (-3%) | 86 (+86%) |
| Afforested_NL_highMono_highIso | 41 (+22%) | 74 (+60%) |

To isolate the impact of afforestation, the BVOC emissions changes presented in Table 5 are all calculated from pairs of simulations with elevated atmospheric $CO_2$ concentration. It should be noted that in the absence of any additional woodlands in our simulations, $CO_2$ inhibition leads to a 15% decrease in isoprene emissions from 39 kt yr$^{-1}$ in our *Baseline* experiment to 33 kt yr$^{-1}$ in our *Present_day_Current_UK_mix* experiment. When comparing the emissions from our *Baseline* experiment to our *Afforested_Current_UK_mix* experiment with additional woodland and elevated $CO_2$ (41 kt yr$^{-1}$), we see only a 5% increase in isoprene emissions because the $CO_2$ inhibition effect offsets a portion of the increase in emission due to additional woodlands.

Figure 9 illustrates the relationship between changes in isoprene and total monoterpene emissions across the five afforestation experiments. With the combination of species mixtures examined, we demonstrate the possibility for isoprene emission to be reduced whilst delivering ambitious afforestation by 2050. However, in experiments where isoprene is reduced (by as much as -3%), monoterpenes increase by at least 48%. All experiments result in an increase in monoterpenes. This

reflects the fact that all afforestation scenarios include tree PFTs that are higher emitters of
monoterpenes than the grassland that has been replaced. Our experiment
*Afforested_BL_lowMono_highIso* demonstrates the potential to minimise the increase in
monoterpenes, with an increase of just 5%, however this is estimated to deliver the most extreme
change across all experiments in relation to isoprene, which increases by over 120%. Our experiments
demonstrate the range of futures that net-zero aligned UK afforestation could deliver for BVOC
emission from natural vegetation. We compare our rates of change to estimates of BVOC emissions
from Purser et al. (2023) for the UK. When scaled to the same level of afforestation, their scenarios
estimate changes in isoprene between -4 and +53%, whilst the change in monoterpenes in their
scenarios is between approximately -3 and +42% (Purser et al., 2023). The range of UK tree species
explored in our paper captures greater extremes in emission potentials for both monoterpenes and
isoprene than Purser et al. (2023) and we therefore estimate that more substantial relative changes in
emissions are possible. However, the scale of change also depends on the land cover, and therefore
emissions, that have been replaced, contributing uncertainty to estimates of relative change.

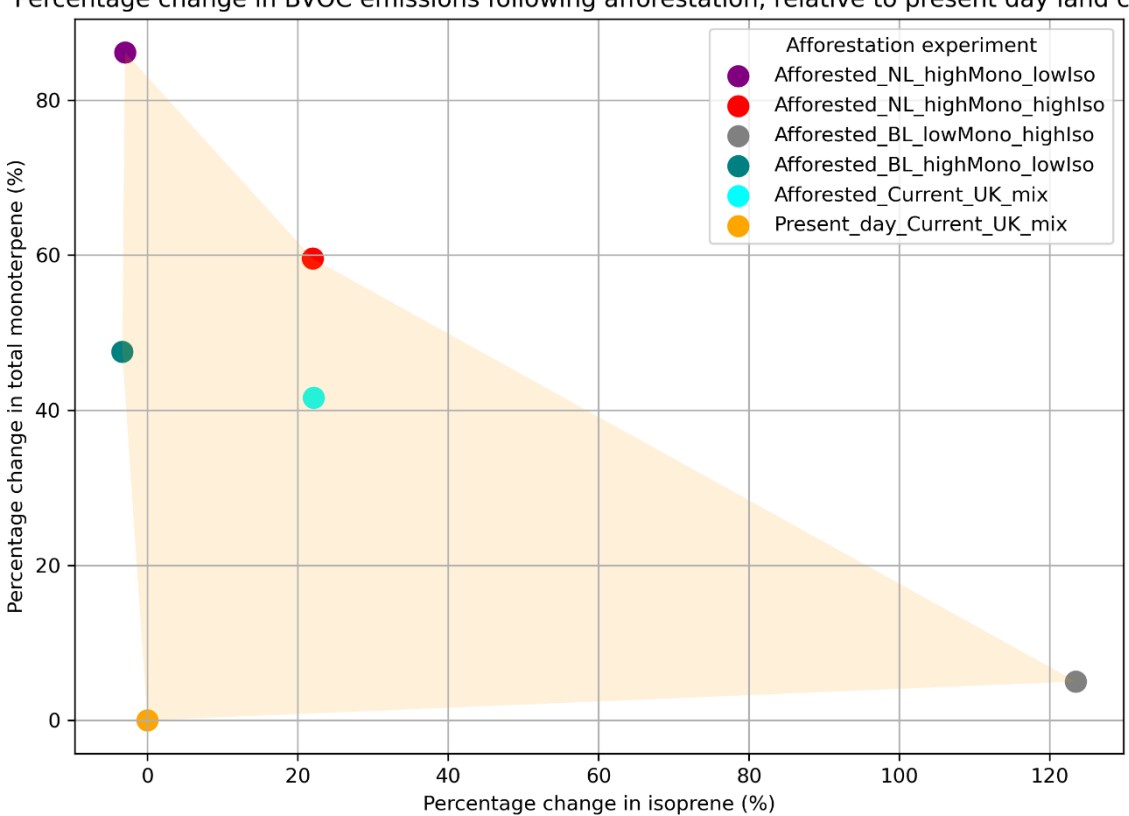

*Figure 9. Relationship between the percentage change in modelled estimates of annual UK isoprene and total monoterpene*
*emissions by afforestation experiment, compared to present day land cover.*

Estimates of BVOC emissions used in this study are limited by assumptions regarding the age of trees
and therefore the calculation of rate of emission in MEGAN (Guenther et al., 1993, 2012). The trees
contributing to the achievement of 19% woodland cover by 2050 would be planted gradually over
decades rather than all being one age. MEGAN makes assumptions about the age of trees based on
information regarding biomass, PFT distribution and leaf age activity factors which inform the values
estimated here (Guenther et al., 1993, 2012). In addition, the specific location of new woodlands which
is unknown is an additional source of uncertainty in BVOC emissions.
Further, estimates of BVOC emissions will be impacted by localised variations, such as tree health, leaf
area index, exposure to stress such as drought, and infestation with tree pests or disease. These
unknowns will further impact the overall change in emissions that are observed in the future from
increased woodland cover. Comparing simulated concentrations from our emissions to observations
would help to understand the ability of the model to capture these sensitivities.
Studies have demonstrated that models, including MEGAN, represent emissions of isoprene during
episodes of drought poorly, both underestimating and over-estimating depending on the severity and
longevity of drought episodes (Jiang et al., 2018; Otu-Larbi et al., 2020; Potosnak et al., 2014; Seco et
al., 2015). For example, using the FORCAsT model Otu-Larbi et al., (2020) demonstrated isoprene
emissions could be underestimated by as much as 40% during a drought episode. Modifications to
account for drought have been found to reduce residuals between simulated and observed emissions
of BVOCs when estimated using MEGAN. Jiang et al. (2018) found isoprene emissions could be as much
as 17% lower when MEGAN was modified to better capture the physiological effects of drought on
emissions and Wang et al. (2022) found that global total isoprene emissions simulated by the CLM
reduced by 10-11% with a representation of water stress based on wilting factors.  We quantify the
sensitivity of our *Baseline*  and *Afforested_Current_UK_mix* experiments to the algorithm of Wang et
al. (2022). In both cases, total annual emissions of isoprene are reduced, by 0.81 kt yr$^{-1}$ (2%;
*Afforested_Current_UK_mix*) and 1.06 kt yr$^{-1}$ (2.74%; *Baseline*) compared to our original estimates.
When we consider how the Wang et al. (2022) algorithm affects the change in emissions due to
afforestation and an increase in atmospheric $CO_2$ concentration (for this particular pair of
experiments), we find that it increases in magnitude from 1.99 kt yr$^{-1}$ (5.14%) to 2.24 kt yr$^{-1}$ (5.95%;
Supplementary Table S2). This suggests that despite the overall impact of the algorithm being a
lowering of isoprene emissions in individual scenarios, the emissions changes we present in Table 5
could all be slightly higher with the updated algorithm.
Two of our experiments (*Afforested_NL_highMono_lowIso* and *Afforested_BL_highMono_lowIso*)
suggest that a reduction in isoprene emissions is possible with an increase in woodland cover of around
50% (or 6% absolute additional woodland cover), when the increase is achieved through planting of
either broadleaf or needleleaf species. This is attributed to the lower isoprene emission factors applied
to tree PFTs in these experiments than those of the land cover that was replaced.

4. Conclusion
This study presents, for the first-time, estimates of BVOC emissions that are consistent with net-zero
aligned afforestation in the UK using tree species suitability information coupled with regionally
appropriate emissions data. Using the CLMv4.5 and MEGANv2.1, we estimate present-day emissions
of isoprene and total monoterpenes, and examine changes to emissions that may occur with around
a 50% increase in woodland cover, delivering absolute woodland cover of approximately 19%. We
present a new estimate for present-day emissions of isoprene at 39 kt yr$^{-1}$, and 46 kt yr$^{-1}$ for
monoterpenes, which is within the range of previous estimates for this region.
Whilst recommendations for afforestation within pathways to net-zero GHG emissions vary between
30,000 and 50,000 ha yr$^{-1}$, this study chooses to investigate the potential impact on BVOCs associated
with the higher rate of planting, equivalent to delivering UK woodland cover of 19% by the year 2050.
This enables us to present an estimate of the potential greatest change in BVOCs relevant to the UK
and representative of the greatest ambition for planting referenced in literature and policy published
to date. The Royal Society report 'Effects of Net-Zero Polices and Climate Change on Air Quality' noted
planting high BVOC emitting tree species as part of net-zero policies as one of several areas of concern
for trade-offs between climate change mitigation and air quality (The Royal Society, 2021). Our
experiments illustrate the potential to minimise the increase of specific BVOC emissions whilst
delivering net-zero aligned afforestation through species selection. Our results show that under
elevated atmospheric $CO_2$ concentration, and in a warmer and drier climate, afforestation leads to
changes in UK emissions of isoprene between -3% and 123%, and between 5% and 86% for total
monoterpene emissions, with the outcome depending on the species mixtures selected for planting.
However, limiting the increase in one type of BVOC (e.g. isoprene) appears to correlate with greater
increase in others (e.g. monoterpenes) as shown in Figure 9. Afforestation in the UK should be
informed by best-available knowledge on the resilience of different tree species to climate change and
pests and diseases, as well as their potential to deliver co-benefits in the landscape. Our study
demonstrates how emissions of BVOCs from the UK's future forests will vary with species selection, by
examining variation within and between broadleaf and needleleaf tree species mixtures.
Our experiments assume large scale conversion of grasslands to woodlands to achieve afforestation
targets. We apply this conversion as a simplified approach to sourcing sufficient land cover for
delivering afforestation to estimate the impact on BVOC emissions, and this should not be used to
infer the suitability of any specific location. We recognise that in reality, afforestation will occur
following decisions around land use and the environmental, social and economic impact of the
changes. The land cover that trees replace ultimately will determine the change in emissions that
occurs; as demonstrated here, a decrease in emissions may be seen if woodland cover replaces a land
cover type with greater potential to emit BVOCs. We show that incorporating regionally appropriate
emissions factors, information about present day abundance of tree species, and the likely role of
different species in the UK's future forests, can substantially alter estimates of emissions. Our study
will support future work to estimate the complex interactions between afforestation and atmospheric
composition, including changes in concentrations of atmospheric pollutants.

## Code availability:

The CESM source code can be obtained from the National Center for Atmospheric Research (NCAR) at https://github.com/ESCOMP/CESM, with documentation at https://escomp.github.io/CESM/versions/cesm2.2/html/introduction.html and https://www.cesm.ucar.edu/. Details of CESM compsets are provided at https://docs.cesm.ucar.edu/models/cesm2/config/compsets.html, and information on the CLM component is available at https://www.cesm.ucar.edu/models/clm.

For information specifically about the Model of Emissions of Gases and Aerosols from Nature (MEGAN) version 2.1, guidance and access to model code are available at https://sites.google.com/uci.edu/bai/megan/data-and-code.

## Data availability:

Currently available data used to generate figures in this study can be downloaded from Zenodo at version DOI: https://doi.org/10.5281/zenodo.16881746. These include the emissions potential data underpinning all experiments presented, total annual emissions estimates, and percentage change data.

The concept DOI: https://doi.org/10.5281/zenodo.16881745, provides access to the most recent versions of the dataset. The spatial emissions data generated from each experiment will be added to the same Zenodo record by 31 January 2026 at the latest. For early access or further information, please contact the corresponding author.

The UKCEH land cover maps are available at https://www.ceh.ac.uk/data/ukceh-land-cover-maps.

## Author contributions

Conceptualization of the study was led by HM, with supervision from CS, SA and PF. HM developed the methodology, curated input data and model configurations and carried out all programming and formal data analysis. BS supported HM with elements of input data preparation. HM prepared the paper with editorial contributions from CS, SA, BS and PF.

## Financial support:

This research has been supported by the Natural Environment Research Council **[grant numbers NE/S007458/1 and NE/S015396/1].**

## Competing interests:

At the time of initial submission, Piers M Forster was the interim Chair of the UK's Climate Change Committee. All other authors have no competing interests to declare.

## Acknowledgements

This study was supported by the United Bank of Carbon, who provide CASE award funding to Hazel Mooney. The authors also acknowledge the contributions of Peter Coleman of the Department for Energy Security and Net Zero to the design of this work, and Alicia Hoffman for editorial contributions.

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
