# Peer review of "Future Forests: estimating biogenic emissions from net-zero aligned"

_EGUsphere, 2024_

## Author Response (AR1)

**Response to reviewer comments and summary of changes for manuscript:**

**"Future Forests: estimating biogenic emissions from net-zero aligned afforestation pathways in the UK" by Mooney et al.**

**Referee #1:**

**Comment 1.1**

This manuscript presents the modelling results of the changes in BVOC emissions from UK afforestation. It is an interesting research topic considering the roles of BVOCs in atmospheric composition and net-zero emission actions. My main struggle with this paper is about the method design.

**Author response to comment 1.1**

We thank the referees for their comments and for taking the time to carefully review the work submitted. We thank the editor for the opportunity to respond to these comments, which enables us add necessary details to justify the approach taken and how it relates to the intention and scope of this manuscript. We have done our best to address the concerns below.  Where we are unable to address the comments directly in our revisions, we note our intentions for responding to the recommendations through ongoing and future work that follows.

**Comment 1.2**

First, the authors chose to estimate BVOC emissions from CLM with the embedded MEGAN model. To fit into the CLM or climate data resolution, the model ran at very coarse spatial resolution; many higher resolution inputs (emission factor, leaf area index, etc) have to be averaged, or species distribution data have to be lumped, which can bring large uncertainties to the emission estimations. There are a lot of higher resolutions of climate data, which are sufficient to run the MEGAN model alone to get much higher-resolution estimations of emissions.

**Author response to comment 1.2**

The resolution of modelling in this study is appropriate for the aim of the paper, which is to estimate the change in the total emission of BVOCs over the UK that would result from afforestation equivalent to planting 50,000 hectares of new woodland every year until 2050. Our study aims to estimate the overall change in emissions from an aspirational 19% UK tree cover, compared to the existing 13% UK tree cover, and to highlight the range of possible emission changes when different species mixtures are planted. Whilst we want to assess the impact of an increase in tree cover on BVOC emissions at national scale, we specifically did not want to prescribe exact locations for the new woodlands as this is a complex ecological and societal issue.

We chose to use the Community Land Model in order to simulate the impacts of afforestation in a more integrated way than can be achieved by running the MEGAN algorithm offline. The use of CLM as a framework for running MEGAN ensures consistency between driving meteorology (temperature, precipitation) and key land surface parameters (e.g. soil moisture) that drive variability in BVOC emissions in MEGAN. The work presented in this paper forms part of a wider body of work that will explore the relationships between environmental conditions and BVOC emissions in the UK, and this requires the use of a land surface model. For this reason, the resolution of our study is aligned with the configurations most suitable for our use of the CLM. Our simulations are preceded by a 30-year spin-up period to ensure that the model reaches a state of equilibrium where the fluxes of carbon, water and energy are balanced, for each land cover configuration. The embedding of MEGANv2.1 in CLM is detailed in Guenther et a., (2012) and Lawrence et al., (2011). This approach means emissions estimations are mapped specifically to the PFT scheme of the CLM.

On species distribution data and perceived lumping: we do not have individual tree species distribution data available for use in the UK, rather we use higher resolution land cover data (the UKCEH land cover map at 1 km x 1 km) to inform the distribution and abundance of Plant Functional Types used by MEGAN and the CLM. Further, we improve the application of CLM and MEGAN to the UK by calculating new emissions factors for PFTs to represent the tree species suitable for the UK (those currently making up woodland cover, as well as those likely to be planted in the future) (as detailed in Section 2.4). The impact of this change can be seen in Figure 7 where we demonstrate the difference in emissions estimates between the default MEGAN set up (panels (c) and (d)) and when using our more UK appropriate emission factors (panels (a) and (b)). In our response to Referee 2 comment 2.4 we provide more detail on how these emissions factors are adapted to reflect the relative abundance of UK tree species.

**Comment 1.3**

I did not get the idea of choosing the CLM model, and there is no clear description of what variables from CLM (if any) were fed into the MEGAN

**Author response to comment 1.3**

We have addressed the first part of this comment in our response to Comment 1.2. Detail has been added to Section 2 at lines 128-131: 'To quantify the emissions of BVOCs from a range of afforestation experiments, we use the Community Land Model v4.5 (CLM) (Oleson et al., 2013), which has the algorithms of the Model of Emissions of Gases and Aerosols from Nature v2.1 (MEGAN) embedded to estimate emissions of BVOCs from the land surface as categorised by the PFT scheme of the CLM (Guenther et al., 1993, 2012)'.

Regarding how MEGAN uses CLM variables, when operating inside the CLM, MEGAN takes information about vegetation type and fraction (PFT), LAI and meteorological variables from the CLM. These are provided as input data when the CLM is operating offline. Within the CLM, these input variables are used to calculate leaf temperature, soil moisture, soil temperature and humidity. The CLM also calculates canopy variables, including the fraction of sunlit vs shaded leaves and the canopy environment coefficient. To calculate BVOC emissions, MEGAN uses leaf temperature, leaf age, soil moisture, solar radiation (for estimating Photosynthetic Photon Flux Density) and $CO_2$ concentrations (which are used to estimate $CO_2$ inhibition in isoprene) as well as the canopy environment coefficient provided by the CLM.  This explanation is reflected in additional corrections made to Section 2 at lines 131-140. Emissions are estimated for 147 BVOCs, with emission factors assigned to 19 compound classes for each of the 16 PFTs used in the CLM. We have added these details to Section 2.4, at lines 276-279: 'Emission factors are combined with information on vegetation type and distribution (represented by PFTs), LAI, solar radiation and atmospheric $CO_2$ concentration and (calculated by the CLM) soil moisture, leaf age, leaf temperature and the canopy environment coefficient for calculating emissions in MEGAN.'.

**Comment 1.4**

I don't think making decisions based on these coarse resolution maps is so informative. With the coarse resolutions and grouped species, it is still unknown where to grow what species (due to grouped species) to increase or decrease certain BVOC emissions, and the resolution is way too coarse to estimate any air quality impacts.

**Author response to comment 1.4**

In lines 157-165, 188-190 and 460-464 we specifically state that the intention of this paper is not to suggest exactly where species should or should not be planted and we acknowledge the broader societal issues around tree planting locations and land cover change overall. Secondly, the purpose of the paper is not to make specific recommendations about where trees should be planted to mitigate any air quality side effects, but to provide a first estimate of the likely scale of change in BVOC emissions-if the levels of woodland creation national governments in the UK are currently aspiring to are realised in 2050.

The spatial resolution of our calculated BVOC emissions is similar to that of emissions datasets commonly used for atmospheric chemistry and air quality modelling simulations (e.g. 0.5 x 0.5° ECLIPSE emissions: https://iiasa.ac.at/models-tools-data/global-emission-fields-of-air-pollutants-and-ghgs). We plan to use these emissions to investigate the impacts of changes in BVOC emission magnitudes on regional ozone and aerosol abundances across the UK, using regional modelling at a similar spatial scale. Our intention is not to resolve fine-scale local-scale air quality changes, since this would be highly dependent on the detailed distribution of tree planting associated with the afforestation scenarios, which is not the focus of our work (see response to Comment 1.2).

Further, we wish to stress that this is a nationwide study, and therefore there are tradeoffs between the spatial coverage and resolution of this work to ensure the overall experiments align with the research questions we wanted to address, which are UK wide.

**Comment 1.5**

Then, when converting grass to trees, the model should consider the removed emissions from grass as well.

**Author response to comment 1.5**

Our modelling framework means that when afforestation occurs, the increased area of forest PFTs in a grid cell is balanced by an equivalent reduction in the area of grass PFTs (for the UK in the CLM these are C3 grass and Arctic C3 grass). We discuss this at lines 374-378 where we comment on the fact that, in some of our afforestation experiments, grass PFTs have higher emission potentials than the forest PFTs we are replacing them with. This is also illustrated in Figures 8 and S3.

**Comment 1.6**

Lastly, it is unclear why the authors chose to run for one future year, i.e., 2050 and even used the meteorological conditions from 2003. The authors' argument for selecting the meteorological condition from 2003 was based on comparing the maximum 1.5 m air temperature. This does not seem correct to me, considering that temperature is not the only factor influencing BVOC emissions. The year-to-year meteorological variations will undoubtedly affect your current estimations; it does not make sense to only look at 1-year outputs.

**Author response to comment 1.6**

The referee is correct, that the justification for using 2003 as the year of meteorological data for these experiments was based on a comparison of observed and projected maximum 1.5 m air temperatures for the UK, and they are correct that temperature is not the only factor influencing BVOC emissions. However, the choice was based on a desire to examine a year which contained a comparable range of temperatures in the UK to those projected for the year 2050, when it is expected the UK government will have achieved the 19% tree cover modelled in our simulations. The year 2003 featured several heatwaves in the UK, including a major heatwave in early August, which has been studied in the literature for its impacts on pollutant formation, such as $O_3$. Using the year 2003 was based both on this opportunity to later consider how the emission of BVOCs during these extremes influences air quality (future work) but also on the alignment of 2003 temperatures with those projected for 2050. Our main objective was to capture the scale of change in BVOC emissions attributed to the increase in forest cover alone, under potential 2050 conditions. Which would be seen in simulations using other years of meteorological data. It is for this reason that only 1 year of output was examined in this study.

**Referee #2**

**Comment 2.1**

The paper by Mooney et al. attempts to quantify future BVOC emissions from different afforestation pathways in the UK. The paper is within the scope of the journal and presents interesting results in terms of air quality implications. However, a number of technical issues need to be addressed prior to potential publication.

**Author response to comment 2.1**

We thank the referees for their comments and for taking the time to carefully review the work submitted. We thank the editor for the opportunity to respond to these comments,

which enables us add necessary details to justify the approach taken and how it relates to the intention and scope of this manuscript. We have done our best to address the concerns below.

Where we are unable to address the comments directly in our revisions, we note our intentions for responding to the recommendations through ongoing and future work that follows.

**Comment 2.2**

Uncertainty analysis. The authors discuss the effect of $CO_2$ in inhibiting isoprene emissions in the future, but completely lack a proper sensitivity study to account for projected heat and drought waves on future BVOC emissions, which in my opinion is a much more relevant factor for future emission scenarios. It has been shown in many recent studies that the peak of isoprene (and possibly other light-dependent emissions) is not correctly represented with the standard CLM parameterisations based on Megan v2.1 (e.g. Jiang et al., 2018: doi: 10.1016/j.atmosenv.2018.01.026, Seco et al., 2015, doi: 10.1111/gcb.12980, Potosnak et al., 2014: doi: 10.1016/j.atmosenv.2013.11.055; Kaser et al, 2022: doi: 10.5194/acp-22-5603-2022; Otu-Larbi et al., 2019: doi: 10.1111/gcb.14963; Wang et al., 2024: doi: 10.1038/s41467-024-49960-0). In particular, heat waves can increase (or decrease) BVOC emissions, depending on their duration, and are an important factor not considered in this study. Failure to account for this effect is a major shortcoming and makes future BVOC projections particularly questionable. The authors should provide an uncertainty assessment by using different parameterisations in their sensitivity runs to estimate the effect on their future projections.

**Author response to comment 2.2**

Thank you for raising this, we ourselves had recognised the need to consider the sensitivity of these estimates to the range of conditions which can impact BVOC emission rates. Namely, we recognise the particular sensitivity to drought and heatwaves as an area requiring further research. We mention in the manuscript our interest in the year 2003 due to the overlap with projected temperatures for 2050, as well as being interesting from the perspective of the historically significant heatwaves experienced in this year. In order to deal with this aspect with an appropriate degree of detail, we have a companion manuscript underway which investigates sensitivities to such variables before, during and after heatwave episodes. This is based on the experiments detailed in the submitted manuscript. Further, it is our intention to evaluate how simulated atmospheric concentrations of isoprene, resulting from our modelled emissions, compare to

observational data; this will provide opportunity to comment on how well MEGAN is capturing emissions from UK trees during these episodes.

We appreciate your suggestion to provide an assessment of the impact that different parameterisations of the influence of drought on BVOCs could have for our estimates of total emissions. Given our study is considering the difference in BVOC emissions for a 2050 world based on changes in levels of tree cover, whist modifications to the drought sensitive algorithm in MEGAN would bring changes to the estimates of absolute emissions (e.g. Potosnak et al., 2014; Seco et al., 2015 and Jiang et al., 2018), the relative change in emissions between afforestation experiments is unlikely to change as much, with meteorology constant between scenarios.

To acknowledge the absence of detail regarding sensitivities, as well as limitations regarding the drought algorithm in the submitted abstract, we have added the following in Section 3.3 at lines 440-455: 'Future work should examine the sensitivity of these projections to extremes such as heatwave and drought episodes, which are expected to increase in frequency in future. Comparing simulated concentrations from our emissions to observations would help to understand the ability of the model to capture these sensitivities. This would aid our understanding of the uncertainty in changes in emissions and how this may vary with climate futures. Studies have demonstrated MEGAN to be unable to reproduce observed emissions of isoprene during episodes of drought, with MEGAN both underestimating and over-estimating isoprene depending on the severity and longevity of drought episodes (e.g. Potosnak et al., 2014; Seco et al., 2015, Jiang et al., 2018; Otu-Larbi et al., 2020). Modifications to account for drought have been found to reduce residuals between simulated and observed emissions of BVOCs when estimated using MEGAN. For example, Jiang et al., (2018) found isoprene emissions could be as much as 17% lower when MEGAN was modified to better capture the physiological effects of drought on emissions. However, Otu-Larbi et al., (2020) demonstrated isoprene emissions could be underestimated by as much as 40% during the drought episode. Whilst changing the drought parameterisation used here would impact the absolute emissions of BVOCs estimated, the change in emissions associated with afforestation should be less sensitive to this.

**Comment 2.3**

Line 127cc: To quantify emissions a number of datasets are used including ambient concentration data. It is not explained further how, and to what extent these observations are used to validate and/or test the model results.

**Author response to comment 2.3**

Apologies that our description of use of these datasets was not clear. Datasets of meteorological data and atmospheric $CO_2$ are used as driving inputs into the CLM model and MEGAN BVOC emission algorithms. We provide more detail on this in our response to Referee comment 1.3. The meteorological dataset is based on reanalysis data (GSWPv3, Dirmeyer et al., 2006), which incorporates measurements to produce an observationally-constrained estimate of the meteorology for the modelled period. $CO_2$ concentration data is also derived from observational data, produced as part of the CMIP6 for the years 1750-2014.

**Comment 2.4**

Model setup: Can the authors be more concrete on the methodology to attribute species-specific emission potentials to PFTs for estimating the average emission potential using tree inventories for actual present day scenarios? The assumptions themselves may be valid, but they are simplifications with associated uncertainties. Quantifying these uncertainties is important to understand how much confidence to place in the conclusions.

**Author response to comment 2.4**

In order to estimate average emissions for present day scenarios, we searched for species-specific emissions potential data for tree species that make up a large proportion of present-day tree cover in the UK.

Information about the proportion of tree species in the UK was obtained from the Forest Research forestry statistics report referenced in line 205 (Forest Research, 2023). Based on this, we sought emissions potential data for 25 tree species from studies where emissions potentials were estimated for the UK, or Europe more generally. Our search returned emissions potential data for 10 broadleaf species and 6 needleleaf trees. For each species a varying number of data points were available, but for each species the mean value was calculated. The error bars shown in Figure 3 and Figure 4 capture the uncertainty of emissions potentials for the different tree species, displaying the minimum and maximum values obtained from the literature. The captions of Figure 3 and Figure 4 have been modified to clarify this. To then apply emissions potentials to the corresponding PFTs, a weighted mean was calculated. Specifically, data for the relative distribution of each of those species at present was used to calculate an average emissions potential for needleleaf and broadleaf PFTs for the UK. The tree species underpinning each are detailed in Table 3.

The application of UK-bespoke estimates for emissions potentials per PFT category are combined with improvements to land surface representation. We detail in section 2.1 the work undertaken to improve representation of UK land cover by regridding the 1 km x 1 km resolution UKCEH land cover map to the 0.47 ° x 0.63 ° resolution of our CLM configuration. This improved the distribution of the Broadleaf Deciduous and Needleleaf Evergreen trees in both boreal and temperate regions. This enables our quantification of the present-day emissions of BVOCs from the land surface. However, we do acknowledge in lines 390-394 the limitation of not also preparing UK-bespoke emissions potentials for non-tree PFTs.

**Summary of changes:**

The following changes to the manuscript have been made to address some of the comments above:

- Lines 128-131 (to address comment 1.3 and comment 1.4) - *'To quantify the emissions of BVOCs from a range of afforestation experiments, we use the Community Land Model v4.5 (CLM) (Oleson et al., 2013), which has the algorithms of the Model of Emissions of Gases and Aerosols from Nature v2.1 (MEGAN) embedded to estimate emissions of BVOCs from the land surface as categorised by the PFT scheme of the CLM (Guenther et al., 1993, 2012)'.*
- Lines 131-140 (to address comment 1.2 and 1.3) - *'When operating inside the CLM, MEGAN takes information about vegetation type and fraction (PFT), LAI and meteorological variables from the CLM. These are provided as input data when the CLM is operating offline. Within the CLM, these input variables are used to calculate leaf temperature, soil moisture, soil temperature and humidity. The CLM also calculates canopy variables, including the fraction of sunlit vs shaded leaves and the canopy environment coefficient. To calculate BVOC emissions, MEGAN uses leaf temperature, soil moisture, soil temperature, solar radiation and $CO_2$ concentrations (which are used to estimate $CO_2$ inhibition in isoprene) as well as the canopy environment coefficient provided by the CLM. Further, European, compound class specific BVOC emissions potential data and tree species distribution data were required.'*

Caption of Figure 3 and Figure 4 – added *'Error bars indicate the minimum and maximum values obtained from the literature.'*

- Lines 276-279 (to address comment 1.4) - *'Emission factors are combined with information on vegetation type and distribution (represented by PFTs), LAI, solar radiation and atmospheric $CO_2$ concentration and (calculated by the CLM) soil*

moisture, leaf age, leaf temperature and the canopy environment coefficient for calculating emissions in MEGAN'

- Lines 440-455 (to address comment 2.2) - 'Future work should examine the sensitivity of these projections to extremes such as heatwave and drought episodes, which are expected to increase in frequency in future. Comparing simulated concentrations from our emissions to observations would help to understand the ability of the model to capture these sensitivities. This would aid our understanding of the uncertainty in changes in emissions and how this may vary with climate futures. Studies have demonstrated MEGAN to be unable to reproduce observed emissions of isoprene during episodes of drought, with MEGAN both underestimating and over-estimating isoprene depending on the severity and longevity of drought episodes (e.g. Potosnak et al., 2014; Seco et al., 2015, Jiang et al., 2018; Otu-Larbi et al., 2020). Modifications to account for drought have been found to reduce residuals between simulated and observed emissions of BVOCs when estimated using MEGAN. For example, Jiang et al., (2018) found isoprene emissions could be as much as 17% lower when MEGAN was modified to better capture the physiological effects of drought on emissions. However, Otu-Larbi et al., (2020) demonstrated isoprene emissions could be underestimated by as much as 40% during the drought episode. Whilst changing the drought parameterisation used here would impact the absolute emissions of BVOCs estimated, the change in emissions associated with afforestation should be less sensitive to this.'.
- References have been added to reflect these additions, including Jiang et al., (2018 at lines 606-608, Otu-Larbti et al., (2020) at lines 652-655 and Seco et l., (2015) at lines 677-680.

In addition to changes made in response to reviewer comments, there are a number of changes made to the manuscript in response to one particular issue identified by the authors.

During the discussion phase for the submitted manuscript, we identified a mistake in how coastal grid cells were being dealt with by the land mask in our code. This had implications for our estimates of total emissions from each experiment. Emissions totals have been recalculated to account for the appropriate number of grid cells, and the correct fraction of land in these grid cells. The changes to values are minor (<5%) and do not impact the magnitude or direction of the changes reported. There is no impact on the messages presented in this manuscript. We have updated the following figures and lines of text to reflect the corrections detailed above:

- Lines 21-25 – '*Our estimate of current annual UK emissions is 39 kt yr$^{-1}$ for isoprene and 46 kt yr$^{-1}$ for total monoterpenes. Broadleaf afforestation results in a change to UK isoprene emission of between -3 and +123%, and a change to total monoterpene emission of between +5% and +47%. Needleleaf afforestation leads to a change in UK isoprene emission of between -3% and +22%, and a change to total monoterpene emission of between +60% and +86%.*'
- Figure 1 updated
- Line 328-329- '*We present an estimate of present-day UK annual isoprene emission of 39 kt yr$^{-1}$ and total monoterpene emission of 46 kt yr$^{-1}$*'
- Figure 6 updated
- Lines 346-350 - '*Experiment Present_day_Current_UK_mix simulates isoprene emissions lower than our Baseline experiment, at 33 kt yr$^{-1}$, whilst emissions of total monoterpene are increased marginally, at 47 kt yr$^{-1}$. The decline of 10% in isoprene emissions can be attributed to the $CO_2$ inhibition effect on the rate of isoprene emission from trees.*'
- Lines 361-365 - '*Combined, these variations result in the small decrease of 5% in isoprene overall in our experiment (to 33 kt yr$^{-1}$) when emissions data is adapted to reflect UK tree species (Fig. 7).*'
- Figure 7 updated
- Figure 8 updated
- Line 387 - '*total emissions increased by 123%.*'
- Lines 387– 390 - '*In experiments Afforested_BL_highMono_lowIso and Afforested NL_highMono_lowIso, the replacement of grassland with tree cover results in a reduction of isoprene emissions, of -3%, and as much as -50% on an individual grid cell level.*'
- Lines 397-399 - '*The greatest changes are observed in the Afforested_NL_highMono_lowIso scenario where some grid cells experience an increase above 250%, and total monoterpene emissions increased by 86%.*'
- Lines 399 – 403 - '*Experiments Afforested_NL_highMono_highIso, Afforested_BL_highMono_lowIso and Afforested_Current_UK_mix demonstrate moderate increases in monoterpene emissions, between 42 and 60%. The smallest change in monoterpene emissions is observed in our Afforested_BL_lowMono_highIso experiment, where a 5% increase is simulated.*'
- Lines 405-407 - '*Our modelled estimates of isoprene emission for the UK with 19% woodland coverage vary between 32 kt yr$^{-1}$ and 74 kt yr$^{-1}$ across the experiments; modelled estimates of total monoterpene emission are between 49 kt yr$^{-1}$ and 87 kt yr$^{-1}$.*'
- Table 5 values updated

- Lines 417-418 - *'However, in experiments where isoprene is reduced (by as much as -3%), monoterpenes increase by at least 47%.'*
- Lines 418-421 - *'Our experiment Afforested_BL_lowMono_highIso demonstrates the potential to minimise the increase in monoterpenes, with an increase of just 5%, however this is estimated to deliver the most extreme change across all experiments in relation to isoprene, which increases by over 120%.'*
- Figure 9 updated
- Lines 466-468 - *'We present a new estimate for present-day emissions of isoprene at 39 kt yr$^{-1}$, and 46 kt yr$^{-1}$ for monoterpenes'*
- Lines 474-476 - *'Our experiments show that changes in UK emissions of isoprene vary between -3% and 123%, and between 5% and 86% for total monoterpene emissions, with the outcome depending on the species mixtures selected for planting.'*
- Supplementary figure S3 and S4 updated.

Secondly, we have added a sentence to methods section 2.7 regarding the resultant level of afforestation delivered by our experiments, which is equivalent to 18.65% at the resolution presented in this work.

- Lines 317-320 - *'We carry out seven experiments without any additional forest, and five with afforestation aligned with achievement of approximately 19% UK woodland coverage. At this resolution, the resulting level of afforestation in model experiments from which emissions are estimate in equivalent to the achievement of 18.65% woodland cover by the year 2050.'*
- Table 4 has been updated to reflect this, with the 'Land data' column now specifying 18.65% afforestation, rather than 19%.

---

## Author Response (AR2)

**Response to reviewer comments on manuscript: "Future Forests: estimating biogenic emissions from net-zero aligned afforestation pathways in the UK" by Mooney et al.**

**Editors comments:**

Dear authors,

Thank you for submitting your revised manuscript and your responses to the referee comments. After reviewing the revised version, I find that the requested major revisions have not been adequately addressed. In response to the referee comments, you have made only minimal changes - two lines of text and a brief paragraph stating that a sensitivity study will be presented in a follow-up paper. As a result, I have invited a third referee (as the original two were unavailable for a second review), who has also raised concerns about the scientific quality of the manuscript. The evaluations from all referees can be summarized as "fair".

The uncertainty analysis or sensitivity study requested by Referee #2 must be included in this manuscript prior to publication.

Furthermore, Referee #3 questions both the novelty and the limited scope of the study (specifically, the lack of consideration for broader implications on air quality and climate), which restricts its potential to advance the field in a meaningful way.

Consequently, I find that your manuscript still requires major revisions to adequately address these critical comments.

At present, the quality of the manuscript is not sufficient for publication in Biogeosciences.

**Authors response to editor comment:**

Thank you for seeking a third referee's opinion and the opportunity to respond to these comments. We have responded to each comment below; where comments address a similar issue we have combined our response. **All additions are given in italics**. In addition to responding to the Referee and Editor comments, there are a few other tracked changes made where we have made minor edits to the paper for clarity. Please note a few numerical values have been corrected due to a minimal rounding error in some instances of from a calculation for total monoterpenes (this has not impacted the scale of the values presented or the messaging and does not apply to all numbers reported).

We appreciate your request to include the sensitivity study raised by Referee #2 and have expanded our manuscript to include this. This addition is described at lines 326-331 and we discuss its impact at lines 494-512. As Reviewer #2 highlighted, previous studies have explored the impact of different algorithms for capturing the impact of heatwaves and drought on isoprene emissions estimated by BVOC emission models (Jiang et al., 2018; Otu-Larbi et al., 2020; Potosnak et al., 2014; Seco et al., 2015). In our manuscript, the results presented to date are based on the isoprene activity factor for soil moisture of MEGANv2.1 as incorporated into

CLM4.5, based on Guenther et al., (2012). In our previous response to Referee #2 we acknowledged that alternative algorithms for drought sensitivity could lead to changes in our emissions estimates; in this new revision, we quantify the impact of using one of the recently proposed parameterisations.

As outlined in Table R1 (which we have added to our Supplementary Information as Table S2), we have performed a pair of new model simulations , corresponding to our *Baseline* experiment (present-day forest cover with 2003 $CO_2$ concentrations) and our *Afforested_Current_UK_mix* experiment (afforestation with present-day mixture of tree species and 2050 $CO_2$ concentration), using the updated drought stress algorithm described in Wang et al., (2022) which simulates water stress response based on wilting factors and root distribution for different PFTs. This algorithm has been adopted into the most recent release of MEGAN as discussed here https://github.com/ESCOMP/CTSM/pull/2588. We quantify the sensitivity of our annual total isoprene emissions for the UK in both our *Baseline* and *Afforested_Current_UK_mix* experiments to the inclusion of the Wang et al., (2022) algorithm. In both cases, total annual isoprene emissions are reduced, by 2% (*Afforested_Current_UK_mix*) and 2.74% (*Baseline*), compared to our original estimates. When including their updated algorithm in global simulations, Wang et al., (2022) calculate a 10-11% reduction in global annual total isoprene emission, an approximately 5 times greater fractional reduction than we see when looking at the UK only.

Next, we consider how inclusion of the Wang et al., (2022) algorithm affects our emission changes due to afforestation. With the original algorithm of Guenther et al. (2012), isoprene emissions increase by 5.14% due to afforestation. When response to water stress is represented in greater detail, the increase in isoprene emissions due to afforestation is slightly greater at 5.95%. It should be noted that the difference between the *Baseline* and *Afforested_Current_UK_mix* experiments in Table R1 includes the impact of afforestation and an increase in atmospheric $CO_2$ concentration from 375 to 500 ppm, therefore the values in Table R1 may be compared to one another but should not be directly compared to those in Table 5 which include the impact of afforestation only.

*Table R1 - Total annual isoprene emissions (kt) and percentage changes associated with two different algorithms for the isoprene activity factor for soil moisture.*

| Afforestation experiment short name | Total annual isoprene emission (kt yr⁻¹) (by isoprene activity factor algorithm for soil moisture) | | Change due to inclusion of Wang et al., (2022) algorithm (kt yr⁻¹; percentage values given in brackets) |
|---|---|---|---|
| | Original (Guenther et al., 2012) | Drought stress (Wang et al., 2022) | |
| *Baseline* | 38.71 | 37.65 | -1.06 (-2.74%) |
| *Afforested_Current_UK_mix* | 40.70 | 39.89 | -0.81 (-2.00%) |
| **Change due to afforestation and increase in $CO_2$ concentration (kt yr⁻¹; percentage values given in brackets)** | 1.99 (+5.14%) | 2.24 (+5.95%) | |

*Additions in the paper are as follows:*

Lines 326-331: *'Previous studies have explored the impact of drought on BVOC emissions (e.g. Potosnak et al., 2014) and developed parameterisations to improve the representation of this process in MEGAN (e.g. Jiang et al., 2018; Wang et al., 2022). To enable quantification of the sensitivity of our results to the representation of the impact of drought on isoprene emissions, extra simulations were run with a modification to the soil moisture activity factor algorithm following Wang et al., (2022), see Supplementary Table S2.'*

Lines 494-512: *'Studies have demonstrated that models, including MEGAN, represent emissions of isoprene during episodes of drought poorly, both underestimating and over-estimating depending on the severity and longevity of drought episodes (Jiang et al., 2018; Otu-Larbi et al., 2020; Potosnak et al., 2014; Seco et al., 2015). For example, using the FORCAsT model Otu-Larbi et al., (2020) demonstrated isoprene emissions could be underestimated by as much as 40% during a drought episode. Modifications to account for drought have been found to reduce residuals between simulated and observed emissions of BVOCs when estimated using MEGAN. Jiang et al., (2018) found isoprene emissions could be as much as 17% lower when MEGAN was modified to better capture the physiological effects of drought on emissions and Wang et al., (2022) found that global total isoprene emissions simulated by the CLM reduced by 10-11% with a representation of water stress based on wilting factors. We quantify the sensitivity of our Baseline and Afforested_Current_UK_mix experiments to the algorithm of Wang et al., ( 2022). In both cases, total annual emissions of isoprene are reduced, by 0.81 kt yr$^{-1}$ (2%; Afforested_Current_UK_mix) and 1.06 kt yr$^{-1}$ (2.74%; Baseline) compared to our original estimates. When we consider how the Wang et al., (2022) algorithm affects the change in emissions due to afforestation and an increase in atmospheric $CO_2$ concentration (for this particular pair of experiments), we find that it increases in magnitude from 1.99 kt yr$^{-1}$ (5.14%) to 2.24 kt yr$^{-1}$ (5.95%; Supplementary Table S2). This suggests that despite the overall impact of the algorithm being a lowering of isoprene emissions in individual scenarios, the emissions changes we present in Table 5 could all be slightly higher with the updated algorithm.'*

**Referees' comments:**

**Referee #3:**

**Comment 3.1**

This manuscript explores possible future scenarios of both increasing woodland cover in the UK from 13% to 17-19% and the impacts that may have on future BVOC emissions under current climate and potential future climate (predictions of climate in 2050) scenarios using the Community Land Model (CLM) (v4.5) with the embedded MEGAN model.

**Author response to comment 3.1**

We thank the referee for their comments and for taking the time to carefully review the work submitted. We thank the editor for the opportunity to respond to these comments, which enables us to add necessary details to justify the approach taken and how it relates to the intention and scope of this manuscript. We have done our best to address the concerns below. Where we are unable to address the comments directly in our revisions, we note our intentions for responding to the recommendations through ongoing and future work that follows. In some cases our responses to comments have been combined.

**Comments 3.2 and 3.6**

3:2: It is my opinion that while the use of regionally appropriate data and realistic woodland planting scenarios for estimating BVOC emissions is commendable and much needed, the manuscript's exclusive focus on modelled BVOC outputs, without examining their implications for air quality or climate, may have limited is scientific impact.

3.6: However, incorporating analysis of how these more regionally specific BVOC emissions, created landcover types and the more realistic planting scenarios could potentially impact air quality and/or climate in the UK, as suggested in the manuscript's opening introduction and concluding sentence, would substantially strengthen its contribution as it would enable a better understanding to how the magnitude of BVOC changes due to increased woodland cover, as described in the manuscript, may impact the UK more widely. Overall, I feel the manuscript would benefit from further development and a substantial revision to meet the publication standards of the journal.

*Author's response to comments 3.2 and 3.6*

We thank the reviewer for these comments and the opportunity to clarity our intentions for this paper, which are to examine the potential future BVOC emissions in the UK associated with afforestation. As the reviewer commends, the paper does this with depth by preparing regionally appropriate BVOC emissions data and making modifications to the default model set up to improve the baseline estimate of BVOC emissions for the UK, this supports our aim to present policy relevant estimates for BVOC emissions associated with afforestation. As well as going into detail to develop the emissions scenarios, we explore the potential distribution of the additional trees (see section 2.2). Emissions potentials used in our scenarios do not just represent UK tree species for which data is available but considers the representation of tree species deemed suitable and feasible to plant in the future, based on climate change and potential risk of pests and disease, as discussed in section 2.3.

We feel that to expand this paper to include air quality and climate assessment would detract from the message of this manuscript and the aims of this work. There is a substantial body of work on vegetation emissions of BVOCs in their own right, which we are contributing to here. We are currently carrying out air quality modelling simulations using the Community Multiscale Air Quality Modelling System (CMAQ), informed by the emissions scenarios presented in this manuscript. This work involves a different model and several datasets not used in this manuscript and as such the write up of this work will require methodological details on modelling and additional literature on chemical mechanisms and air quality policy, which we believe would be inappropriate to try to include in the same paper. Doing so would require a substantial amount of existing content to be reduced, diluting the details of the extensive work that has gone into the results currently presented.

Please also refer to our response to comments 3.3 and 3.20 regarding the scientific advancements of this work relative to previous studies.

**Comments 3.3 and 3.20**

3:3: As BVOC emissions are only one component of a broader assessment, I suggest that the study does not significantly advance beyond previous research, especially considering the substantial uncertainties associated with how the BVOC emission plant functional types are

derived in this or any modelling study, and including the additional uncertainties that underpin the measurement studies that these emission potentials are often based on.

3.20: Line 260-269 – In this section the emphasis is on categorising emission scenarios based on low or high emission of isoprene relative to each other and it indicates tree species that these are representative of as a single tree species. However, in the introduction a lot of the emphasis was based on this study focusing on mixed woodland types with emissions potentials for the woodland that was more representative for UK woodland types. Given this generalisation of high/low isoprene/monoterpene scenarios how is this work an advancement on the previous work of Purser et al. (2023) which used a similar high/low isoprene/monoterpene scenarios although based on a single tree species? Is the overall effect that it produces not a similar in this regard? Could it be that the emission potentials used in this study for future woodland increases still be as uncertain as any emission potentials derived from a single specie?

*Author response to comments 3.3 and 3.20*

We thank the reviewer for considering this aspect of the work in depth and we agree that uncertainties in BVOC emissions remain, particularly due to the grouping of species by plant functional type and limited availability of directly measured emissions potentials. It is important to stress that we are not necessarily trying to reduce uncertainty in emissions estimates, rather we are trying to represent emissions from the UK's woodlands as appropriately as possible to provide information that will be useful for policy and decision makers seeking to build the evidence base around the impacts of extensive afforestation in the UK. To this end, we provide a new estimate of present-day BVOC emissions from the UK (our *Baseline* experiment: isoprene 39 kt yr$^{-1}$ and total monoterpenes 46 kt yr$^{-1}$), of which there are few existing values that cover a relatively large range (i.e., 8 - 110 kt yr$^{-1}$ for isoprene and 31 - 145 kt yr$^{-1}$ for monoterpenes). Rather than being based on default emission potentials for temperate forests, our estimate is informed by the species mixture currently present in the UK. We also consider our afforestation scenarios in the context of higher atmospheric $CO_2$ concentration, which has not been considered before for the UK.

In their 2021 report 'Effects of Net Zero Policies and Climate Change on Air Quality', the Royal Society highlight the planting of high BVOC emitting species and the potential consequences of this on pollutant concentrations as a core example of the potential trade-offs between net-zero policies and air quality (The Royal Society, 2021). The Royal Society discuss the need for consideration of the BVOC emissions of tree species when tree planting to mitigate potential air quality side effects. Our paper seeks to address this by exploring the emissions potentials of UK tree species that make up a high proportion of current woodlands and are likely to feature heavily in future afforestation (Figures 3 and 4) and then evaluating the quantity of emissions delivered by different mixtures of tree species in net-zero aligned afforestation by the year 2050. Our study highlights the varying levels of increase (and in some cases, decrease) in isoprene and monoterpene emissions associated with changing combinations of tree species used to deliver 19% tree cover by 2050. The work provides additional information for policy makers that can help to limit unintended impacts on air quality, by highlighting possible pathways that achieve the required level of afforestation without a huge increase in isoprene emissions. We have added details regarding this at lines 28-30 and 532-536.

'*However, the results highlight possible pathways to achieving 19% woodland cover without requiring large increases in isoprene emissions. The emissions estimates presented provide the opportunity to quantify future impacts on air pollution associated with changes in biogenic*

*emissions, as well as how these impacts would be affected by concurrent changes in anthropogenic emissions.'*

*'The Royal Society report 'Effects of Net-Zero Polices and Climate Change on Air Quality' noted planting high BVOC emitting tree species as part of net-zero policies as one of several areas of concern for trade-offs between climate change mitigation and air quality (The Royal Society, 2021). Our experiments illustrate the potential to minimise the increase of specific BVOC emissions whilst delivering net-zero aligned afforestation through species selection.'*

In the introduction, our mention of mixed woodland types and emissions potentials representative of the UK's woodlands relates to improving our understanding of present-day emissions. We do not mean emissions potentials derived specifically from mixed-woodlands (as opposed to single-species stands for example), we mean mixed in the sense that our study combines emissions potentials of a range of species that are already present in UK woodlands, and likely to feature in future afforestation, covering both needleleaf and broadleaf trees. Our estimate of present-day emissions is based on a weighted mean of the emissions potentials for 16 of the most dominant trees in the UK (based on their relative abundance), with the remaining % of forest not made up by these species represented by a mean emissions value for UK species; this estimate is used as the baseline from which we calculate the change estimated by the year 2050 from the afforested scenarios. To clarify this, we have added the following text at lines 139-141:

*'Our study seeks to explore the potential range of BVOC emissions resulting from a net-zero consistent level of afforestation, if the afforestation was to occur with feasible but extreme (in terms of BVOC emission potentials) mixtures of tree species'.*

Although there are similarities of our work to the research presented in Purser et al., (2023) and we acknowledge this work as a positive contribution to our understanding of the emission behaviour of UK tree species and consequences for air quality associated with UK afforestation, we note several distinctions which make the contributions of our manuscript an advancement to this subject area.

Firstly, the core focus of Purser et al., (2023) is to understand the emissions associated with bioenergy plantations and short rotation forestry, and to quantify the emissions associated with the maximum potential area for bioenergy plantations in the UK. As a result of this, the work focused on 4 tree species suitable for this land cover change, namely Eucalyptus, Aspen, Alder and Sitka Spruce. Our species groupings (used to produce our emissions scenarios) involve 6 needleleaf evergreen trees and 10 broadleaf deciduous trees.

As the reviewer correctly highlights, the species used in Purser et al, (2023) represent a range of emissions tendencies in terms of high/low isoprene/monoterpene emitters. We can certainly understand that our approach could be perceived to achieve the same result conceptually but there are some key differences. The evergreen needleleaf species used in Purser et al., (2023), Sitka Spruce, would correspond to the high isoprene emitting trees of this PFT represented by *NL_highMono_ highIso* in our study (Figure 4), but of the 4 species examined by Purser et al., (2023) there is no low isoprene emitting evergreen needleleaf species which we found to be true of several species in the UK and represented in our *NL_highMono_lowIso* scenario (see Figure 4).

The broadleaf deciduous trees used in Purser et al., (2023) are Alder and Aspen.  Our *BL_highMono_lowIso* category represents much higher monoterpene emission for broadleaf

deciduous trees than is true of either broadleaf species in Purser et al., (2023). The highest isoprene emitter included in Purser et al., (2023) is Aspen, with an emission potential estimated at 22.8 µgC gDW$^{-1}$ hr$^{-1}$. This is less than half the emission potential used in our *BL_lowMono_highIso* scenario which uses emissions potentials for Oak trees. Oak trees represent approximately 8% of present-day UK tree cover (Forest Research, 2023) so it is important that these substantial emitters of isoprene are appropriately captured in estimates of both present day and future emissions.

Despite the generalisation of tree species groups in our paper, the emphasis on extremes of emissions for different categories of trees representative of UK woodland makes the effect very different to the work of Purser et al., (2023) where the focus on tree species suitable for bioenergy. We have added the following text to lines 475-477 where we compare our results to those from Purser et al., (2023):

*'The range of UK tree species explored in our paper captures greater extremes in emission potentials for both monoterpenes and isoprene than Purser et al., (2023) and we therefore estimate that more substantial relative changes in emissions are possible.'*

Finally, our estimates of emissions from future afforestation are simulated under elevated $CO_2$ projected for the year 2050. This is the first study for the UK to consider this.

**Comment 3.4**

It is no surprise here that changing the landcover type to be a higher or lower emitter of isoprene or monoterpene results in a high modelled emission of isoprene or monoterpene for example relative to a substitution of the grass landcover type in the baseline runs in the model. To what extend do these BVOC emissions matter?

*Authors response to comment 3.4*

We agree with the reviewer that it is not a surprise to find higher or lower emissions relative to the grassland that has been substituted. We comment on this in order to highlight how the changes vary between scenarios. Our study illustrates a wide range of changes in monoterpene and isoprene emissions that all occur with the same level of future tree cover and shows opportunities to minimise the increase in certain BVOC emissions whilst delivering 19% tree cover. The extent to which these BVOC emissions matter depends not just on the emissions of trees but the loss of emissions from the land cover which is replaced. We illustrate two cases where the mixtures of tree species represented are lower emitters of isoprene than the land that was replaced (grassland) (see lines 422-425). The potential change in future emissions, and the associated impacts on air quality due to precursor concentrations, depends on the land that is replaced. There is uncertainty therefore in the future changes in emissions of BVOCs associated with afforestation attributed to the variation in BVOC emissions from other land cover types. For example, our study did not replace any cropland with trees.

We had added the following text at lines 429-431:

*'The potential change in BVOC emissions from afforestation would vary with the type of land cover replaced, bringing some uncertainty to the direction of change in emissions in the future.'*

**Comment 3.5 and 3.21**

3.5: In addition to these points, the uncertainties in the location of the woodland to be created in the UK, although I acknowledge that the latest informed current and best-informed efforts by

the authors has been applied to try to overcome this, which is again commendable, it still cannot be overlooked that this is an additional uncertainty in the BVOC emissions.

3.21: I also wonder if a UK averaged tree species mix is useful in this method, given that the distribution of tree species is likely more regional. For example the tree species mix appropriate for the climate of north west Scotland may be vastly different in comparison to south east England. How can we be sure that this tree species mix is suitable UK wide to understand the changes in increase or decrease in BVOC emissions brought about by future increases in woodland? Is there likely to be some uncertainty here if tree species suitability is not taken into account?

*Author response to comments 3.5 and 3.21*

We agree that the exact location of new woodlands will affect their BVOC emissions. To reduce uncertainty in emissions associated with the location of woodland would most likely require a study that does not cover the whole of the UK to enable inclusion of higher resolution information about existing and additional tree cover. Consideration of more specifically where new woodlands would be created requires extremely detailed information about land suitability, availability and socio-economic dimensions which is not the focus of this study. To satisfactorily assess suitability for woodland creation across the UK would be a large undertaking requiring interdisciplinary teams, practitioners and participatory research which is beyond the scope of this modelling work. We agree that this is a limitation that should be highlighted and have added text to our manuscript at lines 193-198 and 487-488 to reiterate this source of uncertainty:

*'Whilst a lack of specific locations for future additional woodland adds extra uncertainty to any estimates of future BVOC emissions, we seek to minimise this by distributing the new woodland in accordance with historical planting rates and future planting commitments across the four nations of the UK.'*

*'In addition, the specific location of new woodlands, which is unknown, is an additional source of uncertainty in BVOC emissions.'*

We agree that our UK averaged mixture is a simplification and does not capture regional variation in the most abundant / suitable species. A challenge to accurately representing present-day BVOC emissions for the UK is the lack of information about species distribution across the countries. We agree that a higher resolution assessment of BVOC emissions based on present-day abundance / future tree species suitability would be a very valuable contribution but unfortunately is beyond the scope of the work we present here.

**Comment 3.7**

I have made additional suggestions for several moderate to minor corrections below, some of which feed into the overall opinion detailed above.

Lines 11- 13 "Woodlands also have the potential to degrade air quality, due to the emission of biogenic 11 volatile organic compounds (BVOCs) which are precursors to major atmospheric pollutants, ozone 12 ($O_3$) and particulate matter (PM)"

Please rephrase this statement as the emphasis is on the woodlands degrading air quality when in fact woodlands may be a benefit on air quality too. I suggest something along the lines of Woodlands emit VOCs....which are known precursors to pollutants and its these which degrade air quality.

Thank you for highlighting this, we agree with your point and have modified the text as follows at lines 11-14:

'*Woodlands can also have benefits for air quality. However, they emit biogenic volatile organic compounds (BVOCs) which are precursors to atmospheric pollutants, such as ozone ($O_3$) and particulate matter (PM), which have the potential to degrade air quality.*'

**Comment 3.8**

Lines 26-27 " Our study highlights the potential for net-zero aligned afforestation to have substantial impacts on UK BVOC emissions, and therefore air quality, but also demonstrates routes to minimizing these impacts through consideration of the emissions potentials of tree species planted."

I suggest given that the year for which the simulations were run was a heatwave and subsequent drought year for the UK and the meteorology (temperature/sunshine hours) and soil moisture may reflect this which has an influence on the BVOC emissions I would specify this in the abstract to make clear to the reader the focus of the manuscript by highlighting this specific example relates to the suggested potential for BVOCs in a warmer and drier, potentially likely future climate effects. This is especially true given the emphasis on elevated CO2 levels too.

In addition, just because BVOC emissions have increased significantly does not mean that in future climates that air quality will be significantly or adversely affected. The correlation is more complex given the future climates could see a reduction in other anthropogenic precursor compounds related to air pollutants.

*Author response to comment 3.8*

We agree and thank the reviewer for highlighting the need to clarify both in the abstract. The following lines have been modified to reflect this:

Lines 18-21: '*Experiments were designed to explore the impact of variation in BVOC emissions potentials between and within plant functional types (PFTs) on estimates of BVOC emissions from UK land cover, in a future warmer climate under elevated atmospheric $CO_2$ concentrations, to understand the scale of change associated with afforestation to 19% woodland cover by the year 2050.*'

Lines 27-32: 'Our study highlights the potential for net-zero aligned afforestation, in a likely warmer and drier future UK climate, to have substantial impacts on BVOC emissions. However, the results highlight possible pathways to achieving 19% woodland cover without requiring large increases in isoprene emissions. The emissions estimates presented here provide the opportunity to quantify future impacts on air pollution associated with changes in biogenic emissions, as well as how these impacts would be affected by concurrent changes in anthropogenic emissions.

We also add detail later in the manuscript to express the uncertainty regarding how BVOCs will impact air quality depending on changes in anthropogenic emissions. Lines 144-148 emphasise the need for this consideration (which we will examine in our next modelling study):

*'The potential impact that increased emissions of BVOCs could have for air quality (associated with their role as precursors to atmospheric pollutants) will depend also on the future of anthropogenic emissions and other precursors, such as $NO_x$, which influence the formation of $O_3$. Our estimates of BVOC emissions from future afforestation scenarios will facilitate this assessment of future air quality impacts.'*

**Comment 3.9:**

Line 30 – I would argue that the pathways are still at this point "suggested pathways" for the mitigation of climate change and just one of several some of which may or may not involve trees.

*Author's response to comment 3.9*

Thank you for highlighting this, you make a good point that these are indeed suggestions and that the role of the terrestrial biosphere varies substantially between different pathways. We have added the word 'suggested' to this sentence in the manuscript (now line 34) to clarify the uncertainty in the exact role of the terrestrial biosphere in the future.

**Comment 3.10**

Line 46-52 – There have also been other suggested benefits of tree planting in the UK which include a reduction in PM 2.5. To give the manuscript a balanced perspective this point should be added to the list in lines 46-52. Please note the following publications that discuss this:

1. Effects of Vegetation on Urban Air Pollution (2018), Air Quality Expert Group Report, DEFRA. https://uk-air.defra.gov.uk/assets/documents/reports/cat09/1807251306_180509_Effects_of_vegetation_on_urban_air_pollution_v12_final.pdf

2. E. Nemitz et al., "Potential and limitation of air pollution mitigation by vegetation and uncertainties of deposition-based evaluations," Philosophical Transactions of the Royal Society A: Mathematical, Physical and Engineering Sciences, vol. 378, no. 2183, Oct. 2020.

3. Purser, G., Heal, M. R., Carnell, E. J., Bathgate, S., Drewer, J., Morison, J. I. L., and Vieno, M.: Simulating impacts on UK air quality from net-zero forest planting scenarios, Atmos. Chem. Phys., 23, 13713–13733, https://doi.org/10.5194/acp-23-13713-2023, 2023.

*Author response to comment 3.10*

Thank you for pointing this out. This was an oversight for deposition to be missing from the list of potential benefits and we have rectified this at lines 50-58 as follows:

*'Increasing forest cover to such an extent could not only bring benefits for climate change mitigation but also a series of co-benefits including habitat creation, flood risk reduction, improving access for people to trees and woodlands (with the associated economic and health benefits), reducing concentrations of particulate matter through deposition, and local temperature reductions (Bolund and Hunhammar, 1999; Costanza et al., 1997; D'Alessandro et al., 2015; Department for Environment, Food and Rural Affairs, 2018; Monger et al., 2022; Nowak, 2022; Purser et al., 2023; Wang et al., 2023). There is also the potential for side-effects and trade-offs, such as the degradation of air quality potentially associated with the emission of*

*biogenic volatile organic compounds (BVOCs) (e.g. Chameides et al., 1988; Churkina et al., 2017; Gu et al., 2021; Rasmussen, 1972).'*

**Comment 3.11**

Line 50-52 - "There is also the potential for delivery of trade-offs, such as the degradation of air quality potentially associated with the emission of biogenic volatile organic compounds (BVOCs)."
Please add references for this statement.

*Authors response to comment 3.11*

Thank you for highlighting this omission, we have added the following references at lines 58:

Chameides, W. L., Lindsay, R. W., Richardson, J., and Kiang, C. S.: The Role of Biogenic Hydrocarbons in Urban Photochemical Smog: Atlanta as a Case Study, Science, 241, 1473–1475, https://doi.org/10.1126/science.3420404, 1988.

Churkina, G., Kuik, F., Bonn, B., Lauer, A., Grote, R., Tomiak, K., and Butler, T. M.: Effect of VOC Emissions from Vegetation on Air Quality in Berlin during a Heatwave, Environ. Sci. Technol., 51, 6120–6130, https://doi.org/10.1021/acs.est.6b06514, 2017.

Gu, S., Guenther, A., and Faiola, C.: Effects of Anthropogenic and Biogenic Volatile Organic Compounds on Los Angeles Air Quality, Environ. Sci. Technol., 55, 12191–12201, https://doi.org/10.1021/acs.est.1c01481, 2021.

Rasmussen, R. A.: What do the hydrocarbons from trees contribute to air pollution, J. Air Pollut. Control Assoc., 22, 537–543, https://doi.org/10.1080/00022470.1972.10469676, 1972.

**Comment 3.12**

Line 59 - Trees' emissions of BVOCs are controlled largely by temperature and light, but also leaf age, atmospheric $CO_2$ concentrations and soil moisture (Guenther et al., 1993; Potosnak et al., 2014; 60 Sharkey, 1996; Zeng et al., 2023).
I suggest rewording for clarity to "BVOC emissions from trees are controlled..... "

*Author response to comment 3.12*
We have updated the wording as suggested at line 65.

**Comment 3.13**

Line 63 – Please add a reference for how plant disease increase may impact quantity and composition change to VOCs.

*Author response to comment 3.13*
Examples of studies that illustrate the impact of tree pests and disease on both the quantity and composition of BVOCs include Irmisch et al., (2014) who observed mixtures of elevated herbivore induced emissions from damaged leaves of poplar trees and found a dominance of terpenoid emissions. Jaakola et al., (2023) explored the change in composition and quantities of BVOC emissions from Norway spruce following attacks of Ips typographus. Emissions were up

to 700x greater in infested trees than in healthy trees, and with more than double the number of different monoterpenes observed in emissions from infested trees than healthy trees. The work of Trowbridge and Stoy (2013) illustrates the tendency for plants to mediate the composition of herbivore-induced BVOC emission to serve as a defence against further herbivory. We have added these references to support lines 69-70.

**Comments 3.14 and 3.15**

3.14: Line 73 – "The emission of monoterpenes is also affected by temperature and light, but the storage of monoterpenes within plants causes distinct patterns in emissions compared to other BVOCs. The emission of BVOCs varies through the day, with a peak in the daytime when incoming solar radiation is at its highest. However, the storage of monoterpenes means their emission is less variable with light."
I feel this section could benefit from some references and a little more clarity around "de novo" synthesis and release of monoterpenes and those which are emitted from storage pools.

3.15: Only some monoterpene emissions have been correlated to both changes in light and temperature and this could be more clearly explained. In addition, I would suggest it is worth making a note of how these daytime releases may vary in light of future climates (heat stress and drought impacted) given this is the focus of the manuscript too.

A suggested publication : Byron, J., Kreuzwieser, J., Purser, G. et al. Chiral monoterpenes reveal forest emission mechanisms and drought responses. Nature 609, 307–312 (2022). https://doi.org/10.1038/s41586-022-05020-5

*Author's response to comments 3.14 and 3.15*

We thank the reviewer for highlighting that this could have been clearer in our manuscript. We have added detail and references to our descriptions of de novo and storage pools of monoterpene emissions. Additions can be found at lines 80-87 as per below.

'*The emission of monoterpenes has historically been thought to be dependent mostly on temperature (Guenther al., 1993), but increasingly evidence shows a light dependence in monoterpene emission for some plant species (e.g. Bao et al., 2008; Ghirardo et al., 2010; Jardine et al., 2015). Some plants have a store of monoterpenes (referred to as pool emissions), which are produced and stored within the leaf until temperatures are high enough for the liquid compound to volatilise (Tingey et al., 1980). De novo emissions are those synthesised in response to both light and temperature without a period of storage in the plant; due to the dependence on light, emissions rates of de novo emissions tend to follow day light patterns (Ghirardo et al., 2010).*'

To expand on the impact that heat stress and drought could have on the release of monoterpenes during daytime hours we add the following to the manuscript at lines 87-93:

'*The response of monoterpene emissions to climate change is influenced by the behaviour of individual enantiomers. Byron et al. (2022) found that drought affects monoterpene enantiomers differently, depending on both the severity and duration of the drought. Since enantiomers can help distinguish between emissions from de novo synthesis and those from storage pools, differing enantiomeric responses introduce uncertainty in future emission estimates—especially if the balance between de novo and storage-derived emissions shifts.*'

**Comment 3.16**

Line 117 – suggest adding the word "tree" before species in "cover with mixed species afforestation".

*Author's response to comment 3.16*
We have made this revision at line 135.

**Comment 3.17**

Line 159 – "here is inferred from the UKCEH" suggest adding ".......landcover map (1km x 1km resolution)" to the caption of figure 1.

*Author's response to comment 3.17*
Revision made at line 179.

**Comment 3.18**

Line 162-164 – please add a reference(s) for this statement.
" The exact locations for afforestation in the UK are undetermined, though the individual four nations (England, Northern Ireland, Scotland, Wales) have their own ambitions relating to the net-zero aligned planting recommendations."

*Author's response to comment 3.18*
Corresponding references added in lines 184-186.

**Comment 3.19**

Line 172. Please add a reference to this line for the basis of this information highlighted in table 1.

*Author's response to comment 3.19*
Corresponding references added in lines 196-198.

**Comment 3.22**

Lines 387-389 "The greatest increase in isoprene was observed in the experiment Afforested_BL_lowMono_highIso, where individual grid cells experienced up to a 250% increase in emissions, and total emissions increased by 123%."

Firstly, is this increase relative to the current BVOC emissions or the 2003 with future $CO_2$ 500ppm concentration BVOC emissions? I wonder if perhaps this result is no surprise given the scenario is a high isoprene emission potential scenario model run? I wonder if these results are as useful given that we know that the model is designed to replace the grassland land cover type and so either isoprene or monoterpene scenarios? Increases or decreases may result in changes to overall isoprene or monoterpene emissions. Maybe a little more explanation of the impact or significance is needed.

*Author's response to comment 3.22*

We thank the reviewer for highlighting that our explanation of these results needs to be clearer. The percentage changes in annual BVOC emissions reported in our manuscript, namely in sections 3.3 (including Figure 8 and Table 5 and Figure 9) and again in section 4, represent the increase relative to our *Present_day_Current_UK_mix* experiment, which involved $CO_2$ concentrations at 500 ppm. We have chosen to represent this difference, as opposed to the difference compared to the *Baseline* experiment (with 2003 $CO_2$) to isolate the difference attributed specifically to afforestation. Due to the large number of simulations involved in this project, and thus resource demand, afforested scenarios were not run at both $CO_2$ concentrations. We have modified lines 405-407 and 415-417 to clarify this:

*'All experiments used to estimate percentage change in emissions use elevated $CO_2$ as projected for 2050, therefore all percentage changes in emissions are attributed to afforestation alone.'*

*'The percentage change reported here is relative to our Present_day_Current_UK_mix experiment, with the change attributed to the different in forest cover (all experiments using elevated $CO_2$ at 500 ppm).'*

To add some more context around the impacts of afforestation and $CO_2$ concentration changes, we have also added the following text at lines 451-458:

*'To isolate the impact of afforestation, the BVOC emissions changes presented in Table 5 are all calculated from pairs of simulations with elevated atmospheric $CO_2$ concentration. It should be noted that in the absence of any additional woodlands in our simulations, $CO_2$ inhibition leads to a 15% decrease in isoprene emissions from 39 kt yr$^{-1}$ in our Baseline experiment to 33 kt yr$^{-1}$ in our Present_day_Current_UK_mix experiment. When comparing the emissions from our Baseline experiment to our Afforested_Current_UK_mix experiment with additional woodland and elevated $CO_2$ (41 kt yr$^{-1}$), we see only a 5% increase in isoprene emissions because the $CO_2$ inhibition effect offsets a portion of the increase in emission due to additional woodlands.'*

We report the percentage change associated with each afforestation experiment to illustrate the scale of variation in how emissions of BVOCs could change depending on the species mixtures selected. Whilst it is not surprising that a high isoprene scenario would deliver the highest percentage change in isoprene, the figure is useful when compared to other experiments, particularly for illustrating the stark contrast between this experiment and those with lower isoprene emissions potentials. Whilst we illustrate the variation in emissions potentials for the different experiments in Figure 5, the percentage changes in total annual emissions reported in section 3.3 support our interpretation of the scale of change we could see. We discuss the difference in percentage change in emissions from our experiments when compared to Purser et al., (2023) at lines 464-469. We have added a line (477-478) to clarify the differences to be not just associated with emissions potentials/tree species represented, but also the land cover replaced.

*'However, the scale of change also depends on the land cover, and therefore emissions, that have been replaced, contributing uncertainty to estimates of relative change.'*

**Comment 3.23**

Line 420 – "All experiments result in an increase in monoterpenes."
Perhaps an explanation could be added after this sentence to reiterate why this might be the case?

*Author's response to comment 3.23*

An increase in monoterpenes in all experiments illustrates the fact that, overall, the needleleaf and broadleaf PFTs that replace grassland PFTs to deliver afforestation have a greater emission potential for monoterpenes and therefore annual total monoterpene emissions are increased in all cases under this land cover conversion. We have added an explanation at lines 464-466 to reflect this:

*'This reflects the fact that all afforestation scenarios include tree PFTs that are higher emitters of monoterpenes than the grassland that has been replaced.'*

**Comment 3.24**

Line 458 - "Two of our experiments (Afforested_NL_highMono_lowIso and Afforested_BL_highMono_lowIso) suggest that a reduction in isoprene emissions is possible with an increase in woodland cover of around 50% (or 6% absolute additional woodland cover), when the increase is achieved through planting of either broadleaf or needleleaf species."

Could you please give an explanation/suggestion as to why this might be the case here.

*Author's response to comment 3.24*

The explanation for this reduction in isoprene is discussed at lines 436-440 regarding the replacement of land covered by grass PFTs of higher isoprene emission potential with the trees in the *Afforested_NL_highMono_lowIso* and *Afforested_BL_highMono_lowIso* experiments with lower isoprene emissions potentials. We have reiterated this at lines 516-517:

*'This is attributed to the lower isoprene emission factors applied to tree PFTs in these experiments than those of the land cover that was replaced.'*